# A Provably Efficient Model-Free Posterior Sampling Method for Episodic Reinforcement Learning

**Christoph Dann**
Google Research

**Mehryar Mohri**
Google Research and NYU

**Tong Zhang**
Google Research and HKUST

**Julian Zimmert**
Google Research

## Abstract

Thompson Sampling is one of the most effective methods for contextual bandits and has been generalized to posterior sampling for certain MDP settings. However, existing posterior sampling methods for reinforcement learning are limited by being model-based or lack worst-case theoretical guarantees beyond linear MDPs. This paper proposes a new model-free formulation of posterior sampling that applies to more general episodic reinforcement learning problems with theoretical guarantees. We introduce novel proof techniques to show that under suitable conditions, the worst-case regret of our posterior sampling method matches the best known results of optimization based methods. In the linear MDP setting with dimension, the regret of our algorithm scales linearly with the dimension as compared to a quadratic dependence of the existing posterior sampling-based exploration algorithms.

## 1 Introduction

A key challenge in reinforcement learning problems is to balance exploitation and exploration. The goal is to make decisions that are currently expected to yield high reward and that help identify less known but potentially better alternate decisions. In the special case of contextual bandit problems, this trade-off is well understood and one of the most effective and widely used algorithms is Thompson sampling [Thompson, 1933]. Thompson sampling is a Bayesian approach that maintains a posterior distribution of each arm being optimal for the given context. At each round, the algorithm samples an action from this distribution and updates the posterior with the new observation. The popularity of Thompson sampling stems from strong empirical performance [Li et al., 2010], as well as competitive theoretical guarantees in the form of Bayesian [Russo and Van Roy, 2014] and frequentist regret bounds [Kaufmann et al., 2012].

The results in the contextual bandit setting have motivated several adaptations of Thompson sampling to the more challenging Markov decision process (MDP) setting. Most common are model-based adaptations such as PSRL [Strens, 2000, Osband et al., 2013, Agrawal and Jia, 2017] or BOSS [Asmuth et al., 2012], which maintain a posterior distributions over MDP models. These algorithms determine their current policy by sampling a model from this posterior and computing the optimal policy for it. The benefit of maintaining a posterior over models instead of the optimal policy or optimal value function directly is that posterior updates can be more easily derived and algorithms are easier to analyze. However, model-based approaches are limited to small-scale problems where a realizable model class of moderate size is available and where computing the optimal policy of a model is computationally tractable. This rules out many practical problems where the observations are rich, e.g. images or text.

35th Conference on Neural Information Processing Systems (NeurIPS 2021).

There are also model-free posterior sampling algorithms that are inspired by Thompson sampling. These aim to overcome the limitation of model-based algorithms by only requiring a value-function class and possibly weaker assumptions on the MDP model. Several algorithms have been proposed [e.g. Osband et al., 2016a, Fortunato et al., 2017, Osband et al., 2018] with good empirical performance but with no theoretical performance guarantees. A notable exception is the randomized least-squares value iteration (RLSVI) algorithm by Osband et al. [2016b] that admits frequentist regret bounds in tabular [Russo, 2019] and linear Markiov decision processes [Zanette et al., 2020a]. However, to the best of our knowledge, no such results are available beyond the linear setting.

In contrast, there has been impressive recent progress in developing and analyzing provably efficient algorithms for more general problem classes based on the OFU (optimism in the face of uncertainty) principle. These works show that OFU-based algorithm can learn a good policy with small sample-complexity or regret as long as the value-function class and MDP satisfies general structural assumptions. Those assumptions include bounded Bellman rank [Jiang et al., 2017], low inherent Bellman error [Zanette et al., 2020b], small Eluder dimension [Wang et al., 2020] or Bellman-Eluder dimension [Jin et al., 2021]. This raises the question of whether OFU-based algorithms are inherently more suitable for such settings or whether it is possible to achieve similar results with a model-free posterior sampling approach. In this work, we answer this question by analyzing a posterior sampling algorithm that works with a Q-function class and admits worst-case regret guarantees under general structural assumptions. Our main contributions are:

- We derive a *model-free* posterior sampling algorithm for reinforcement learning in general Markov decision processes and value function classes.
- We introduce a new proof technique for analyzing posterior sampling with optimistic priors.
- We prove that this algorithm achieves near-optimal worst-case regret bounds that match the regret of OFU-based algorithms and improve the best known regret bounds for posterior sampling approaches.

## 1.1 Further Related Work

Several extension of TS have been proposed for non-linear function classes in contextual bandits [Zhang et al., 2020, Kveton et al., 2020, Russo and Van Roy, 2014]. For tabular MDPs, Agrawal et al. [2020] improves the regret bounds for RLSVI and Xiong et al. [2021] show that an algorithm with randomized value functions can be optimal. In the same setting, Pacchiano et al. [2020] proposed an optimism based algorithm that uses internal noise, which can be interpreted as a posterior sampling method. Jafarnia-Jahromi et al. [2021] analyzes a posterior sampling algorithm for tabular stochastic shortest path problems. Considering the Bayesian regret, the seminal paper of Osband and Van Roy [2014] provides a general posterior sampling RL method that can be applied to general model classes including linear mixture MDPs.

## 2 Setting and Notation

**Episodic Markov decision process.**  We consider the episodic Markov decision process (MDP) setting where the MDP is defined by the tuple $(\mathcal{X}, \mathcal{A}, H, P, r)$. Here, $\mathcal{X}$ and $\mathcal{A}$ are state and action spaces. The number $H \in \mathbb{N}$ is the length of each episode and $P = \{P^h\}_{h=1}^{H}$ and $R = \{R^h\}_{h=1}^{H}$ are the state transition probability measures and the random rewards. The agent interacts with the MDP in $T$ episodes of length $H$. In each episode $t \in [T] = \{1, 2, \ldots, T\}$, the agent first observes an initial state $x_t^1 \in \mathcal{X}$ and then, for each time step $h \in [H]$, the agent takes an action $a_t^h \in \mathcal{A}$ and receives the next state $x_t^{h+1} \sim P^h(\cdot|x_t^h, a_t^h)$ and reward $r_t^h \sim R^h(\cdot|x_t^h, a_t^h)$. To simplify our exposition, we assume that the initial states $x_t^1 = x^1$ are identical across episodes. When initial states are stochastic, we can simply add a dummy initial state and increase $H$ by 1 to ensure this. Our approach can also be generalized to adversarial initial states which we discuss in the appendix. We denote by $\pi_t \colon \mathcal{X} \times [H] \to \mathcal{A}$ the agent's policy in the $t$-th episode, that is, $a_t^h = \pi_t(x_t^h, h)$. For a given policy $\pi$, the value functions are defined for all $h \in [H]$ and state-action pairs $(x, a)$ as

$$Q_h^\pi(x, a) = r^h(x, a) + \mathbb{E}_{x' \sim P^h(x,a)} \left[ V_{h+1}^\pi(x') \right], \qquad V_h^\pi(x) = Q_h^\pi(x, \pi(x, h)),$$

where $r^h(x, a) = \mathbb{E}_{r^h \sim R^h(x,a)}[r^h] \in [0, 1]$ is the average immediate reward and $V_{H+1}^\pi(x) = 0$ for convenience. We further denote by $\mathcal{T}_h^\star$ the Bellman optimality operator that maps any state-action

function $f$ to

$$[\mathcal{T}_h^\star f](x,a) = r^h(x,a) + \mathbb{E}_{x' \sim P^h(x,a)} \left[ \max_{a' \in \mathcal{A}} f(x',a') \right] .$$

The optimal Q-function is given by $Q_h^\star = \mathcal{T}_h^\star Q_{h+1}^\star$ for all $h \in [H]$ where again $Q_{H+1}^\star = 0$.

**Value function approximation.** We assume the agent is provided with a Q-function class $\mathcal{F} = \mathcal{F}_1 \times \mathcal{F}_2 \times \cdots \times \mathcal{F}_H$ of functions $f = \{f^h\}_{h \in [H]}$ where $f^h \colon \mathcal{X} \times \mathcal{A} \to \mathbb{R}$. For convenience, we also consider $\mathcal{F}_{H+1} = \{\mathbf{0}\}$ which only contains the constant zero function, and we define $f^h(x) = \max_{a \in \mathcal{A}} f^h(x,a)$. Before each episode $t$, our algorithm selects a Q-function $f_t \in \mathcal{F}$ and then picks actions in this episode with the greedy policy $\pi_{f_t}$ w.r.t. this function. That is, $a_t^h = \pi_{f_t}(x_t^h, h) \in \arg\max_{a \in \mathcal{A}} f_t^h(x_t^h, a)$.

We make the following assumptions on the value-function class:

**Assumption 1** (Realizability). *The optimal Q-function is in the class $Q_h^\star \in \mathcal{F}_h$ for all $h \in [H]$.*

**Assumption 2** (Boundedness). *There exists $b \geq 1$ such that $f^h(x,a) \in [0, b-1]$ for all $x,a \in \mathcal{X} \times \mathcal{A}$ and $f = \{f^h\}_{h \in [H]} \in \mathcal{F}$.*

**Assumption 3** (Completeness). *For all $h$ and $f^{h+1} \in \mathcal{F}_{h+1}$, there is a $f^h \in \mathcal{F}_h$ such that $f^h = \mathcal{T}_h^\star f^{h+1}$.*

**Additional notation.** For any $f^h \in \mathcal{F}_h$, we define the short-hand notation $f^h(x) = \max_{a \in \mathcal{A}} f^h(x,a)$ and for any $f \in \mathcal{F}$, $h \in [H]$ and state-action pair $x,a$, we define the *Bellman residual* as

$$\mathcal{E}_h(f; x, a) = \mathcal{E}(f^h, f^{h+1}; x, a) = f^h(x,a) - \mathcal{T}_h^\star f^{h+1}(x,a) .$$

We measure the performance of an algorithm that produces a sequence of policies $\pi_1, \pi_2, \ldots$, by its *regret*

$$\mathrm{Reg}(T) = \sum_{t=1}^{T} (V_1^\star(x^1) - V_1^{\pi_t}(x^1))$$

after $T$ episodes.

## 3  Conditional Posterior Sampling Algorithm

We derive our posterior sampling algorithm by first defining the prior over the function class and our likelihood model of an episode given a value function.

**Optimistic prior.** We assume that the prior $p_0$ over $\mathcal{F}$ has the form

$$p_0(f) \propto \exp(\lambda f^1(x^1)) \prod_{h=1}^{H} p_0^h(f^h) \tag{1}$$

where $p_0^h$ are distributions over each $\mathcal{F}_h$ and $\lambda > 0$ is a parameter. This form assumes that the prior factorizes over time steps and that the prior for the value functions of the first time step prefers large values for the initial state. This optimistic preference helps initial exploration and allows us to achieve frequentist regret bounds. A similar mechanism can be found in existing sampling based algorithms in the form of optimistic value initializations [Osband et al., 2018] or forced initial exploration by default values [Zanette et al., 2020a].

**Temporal difference error likelihood.** Consider a set of observations acquired in $t$ episodes $S_t = \{x_s^h, a_s^h, r_s^h\}_{s \in [t], h \in [H]}$. To formulate the likelihood of these observations, we make use of the squared temporal difference (TD) error, that for time $h$, is

$$L^h(f; S_t) = L^h(f^h, f^{h+1}; S_t) = \sum_{s=1}^{t} (f^h(x_s^h, a_s^h) - r_s^h - f^{h+1}(x_s^{h+1}))^2 .$$

For $h = H$, we can choose $x_s^{H+1}$ arbitrarily since by definition $f^{H+1}(x) = 0$ for all $x$. We now define the likelihood of $S_t$ given a value function $f \in \mathcal{F}$ as

$$p(S_t|f) \propto \prod_{h=1}^{H} \frac{\exp\left(-\eta L^h(f^h, f^{h+1}; S_t)\right)}{\mathbb{E}_{\tilde{f}^h \sim p_0^h} \exp(-\eta L^h(\tilde{f}^h, f^{h+1}; S_t))} \ , \tag{2}$$

where $\eta > 0$ is a parameter. Readers familiar with model-free reinforcement learning methods likely find the use of squared temporal difference error in the numerator natural as it makes transition samples with small TD error more likely. Most popular model free algorithm such as Q-learning [Watkins and Dayan, 1992] rely on the TD error as their loss function. However, the normalization in the denominator of (2) may seem surprising. This term makes those transitions more likely that have small TD error under the specific $f^h, f^{h+1}$ pair compared to pairs $(\tilde{f}, f^{h+1})$ with $\tilde{f}$ is drawn from the prior. Thus, transitions are encouraged to explain the specific choice of $f^h$ for $f^{h+1}$. This normalization is one of our key algorithmic innovations. It allows us to related a small loss $L^h$ to a small bellman error and circumvent the double-sample issue [Baird, 1995, Dann et al., 2014] of the square TD error.

Combining the prior in (1) and likelihood in (2), we obtain the posterior of our algorithm after $t$ episodes

$$p(f|S_t) \propto \exp(\lambda f^1(x^1)) \prod_{h=1}^{H} q(f^h|f^{h+1}, S_t), \tag{3}$$

$$\text{where} \quad q(f^h|f^{h+1}, S_t) = \frac{p_0^h(f^h) \exp\left(-\eta L^h(f^h, f^{h+1}; S_t)\right)}{\mathbb{E}_{\tilde{f}^h \sim p_0^h} \exp(-\eta L^h(\tilde{f}^h, f^{h+1}; S_t))} \ .$$

Note that the conditional probability $q(f^h|f^{h+1}, S_t)$ samples $f^h$ using the data $S_t$ and the TD error $L^h(f^h, f^{h+1}; S_t)$, which mimics the model update process of Q-learning, where we fit the model $f^h$ at each step $h$ to the target computed from $f^{h+1}$. The optimistic prior encourages exploration, which is needed in our analysis. It was argued that such a term is necessary to achieve the optimal frequentist regret bound for Thompson sampling in the bandit case Zhang [2021]. For the same reason, we employ it for the analysis of posterior sampling in episodic RL. The losses $L^h$ will grow with the number of rounds $t$ and once the effect of the first $\exp(\lambda f^1(x^1))$ factor has become small enough, the posterior essentially performs Bayesian least-squares regression of $f^h$ conditioned on $f^{h+1}$. We thus call our algorithm *conditional posterior sampling*. This algorithm, shown in Algorithm 1 simply samples a Q-function $f_t$ from the posterior before the each episode, follows the greedy policy of $f_t$ for one episode and then updates the posterior.

---

**Algorithm 1:** Conditional Posterior Sampling Algorithm

**Input:** value function class $\mathcal{F}$, learning rate $\eta$, prior optimism parameter $\lambda$, number of rounds $T$

1 **for** $t = 1, 2, \ldots, T$ **do**
2     Draw Q-function $f_t \sim p(\cdot|S_{t-1})$ according to posterior (3)
3     Play episode $t$ using the greedy policy $\pi_{f_t}$ and add observations to $S_t$

---

We are not aware of a provably computationally efficient sampling procedure for (3). Since the primary focus of our work is statistical efficiency we leave it as an open problem for future work to investigate the empirical feasibility of approximate samplers or to identify function classes that allow for efficient sampling. Another potential improvement is to derive a conditional sampling rule that allows to sample $(f_t^h)_{h=1}^{H}$ iteratively instead of jointly from a single posterior.

## 4   Regret of Conditional Posterior Sampling

We will now present our main theoretical results for Algorithm 1. As is common with reinforcement learning algorithms that work with general function classes and Markov decision processes, we express our regret bound in terms of two main quantities: the effective size of the value function class $\mathcal{F}$ and a structural complexity measure of the MDP in combination with $\mathcal{F}$.

**Function Class Term.** In machine learning, the gap between the generalization error and the training error can be estimated by an appropriately defined complexity measure of the target function class. The analyses of optimization-based algorithms often assume finite function classes for simplicity and measure their complexity as $|\mathcal{F}|$ [Jiang et al., 2017] or employ some notion of covering number for $\mathcal{F}$ [Wang et al., 2020, Jin et al., 2021]. Since our algorithm is able to employ a prior $p_0$ over $\mathcal{F}$ which allows us to favor certain parts of the function space, our bounds instead depend on the complexity of $\mathcal{F}$ through the following quantity:

**Definition 1.** *For any function $f' \in \mathcal{F}_{h+1}$, we define the set $\mathcal{F}_h(\epsilon, f') = \left\{ f \in \mathcal{F}_h \colon \sup_{x,a} |\mathcal{E}_h(f, f'; x, a)| \leq \epsilon \right\}$ of functions that have small Bellman error with $f'$ for all state-action pairs. Using this set, we define*

$$\kappa(\epsilon) = \sup_{f \in \mathcal{F}} \sum_{h=1}^{H} \ln \frac{1}{p_0^h(\mathcal{F}_h(\epsilon, f^{h+1}))}.$$

The quantity $p_0^h(\mathcal{F}_h(\epsilon, f))$ is the probability assigned by the prior to functions that approximately satisfy the Bellman equation with $f$ in any state-action pair. Thus, the complexity $\kappa(\epsilon)$ is small if the prior is high for any $f$ and $\kappa(\epsilon)$ represents an approximate completeness assumption. In fact, if Assumption 3 holds we expect $\kappa(\epsilon) < \infty$ for all $\epsilon > 0$. In the simplest case where $\mathcal{F}$ is finite, $p_0^h(f) = \frac{1}{|\mathcal{F}_h|}$ is uniform and completeness holds exactly, we have

$$\kappa(\epsilon) \leq \sum_{h=1}^{H} \ln |\mathcal{F}_h| = \ln |\mathcal{F}| \qquad \forall \epsilon \geq 0 \,.$$

For parametric models, where each $f^h = f_\theta^h$ can be parameterized by a $d$-dimensional parameter $\theta \in \Omega_h \subset \mathbb{R}^d$, then a prior $p_0^h(\theta)$ on $\Omega_h$ induces a prior $p_0^h(f)$ on $\mathcal{F}_h(\epsilon, f)$. If $\Omega_h$ is compact, then we can generally assume that the prior satisfies $\sup_\theta \ln \frac{1}{p_0^h(\{\theta' : \|\theta' - \theta\| \leq \epsilon\})} \leq d \ln(c'/\epsilon)$ for an appropriate constant $c' > 0$ that depends on the prior. If further $f^h = f_\theta^h$ is Lipschitz in $\theta$, then we can assume that $\ln \frac{1}{p_0^h(\mathcal{F}_h(\epsilon, f^{h+1}))} \leq c_0 d \ln(c_1/\epsilon)$ for constants $c_0 > 0$ and $c_1 > 0$ that depend on the prior and the Lipschitz constants. This implies the following bound for $d$ dimensional parameteric models

$$\kappa(\epsilon) \leq c_0 H d \ln(c_1/\epsilon). \tag{4}$$

**Structural Complexity Measure** In our regret analysis, we need to investigate the trade-off between exploration and exploitation. The difficulty of exploration is measured by the structure complexity of the MDP. Similar to other complexity measures such as Bellman rank [Jiang et al., 2017], inherent Bellman error [Zanette et al., 2020b] or Bellman-Eluder dimension [Jin et al., 2021], we use a complexity measure that depends on the Bellman residuals of the functions $f \in \mathcal{F}$ in our value function class. Our measure *online decoupling coefficient* quantifies the rate at which the average Bellman residuals can grow in comparison to the cumulative squared Bellman residuals:

**Definition 2** (Online decoupling coefficient). *For a given MDP $M$, value function class $\mathcal{F}$, time horizon $T$ and parameter $\mu \in \mathbb{R}^+$, we define the online decoupling coefficient $\mathrm{dc}(\mathcal{F}, M, T, \mu)$ as the smallest number $K$ so that for any sequence of functions $\{f_t\}_{t \in \mathbb{N}}$ and their greedy policies $\{\pi_{f_t}\}_{t \in \mathbb{N}}$*

$$\sum_{h=1}^{H} \sum_{t=1}^{T} \left[ \mathbb{E}_{\pi_{f_t}}[\mathcal{E}_h(f_t; x^h, a^h)] \right] \leq \mu \sum_{h=1}^{H} \sum_{t=1}^{T} \left[ \sum_{s=1}^{t-1} \mathbb{E}_{\pi_{f_s}}[\mathcal{E}_h(f_t; x^h, a^h)^2] \right] + \frac{K}{4\mu}.$$

The online decoupling coefficient can be bounded for various common settings, including tabular MDPs (by $|\mathcal{X}||\mathcal{A}|HO(\ln T)$) and linear MDPs (by $dHO(\ln T)$ where $d$ is the dimension). We defer a detailed discussion of this measure with examples and its relation with other complexity notions to later.

**Main Regret Bound.** We are now ready to state the main result of our work, which is a frequentist (or worst-case) expected regret bound for Algorithm 1:

**Theorem 1.** *Assume that parameter $\eta \leq 0.4b^{-2}$ is set sufficiently small and that Assumption 2 holds. Then for any $\beta > 0$, the expected regret after $T$ episodes of Algorithm 1 on any MDP $M$ is bounded as*

$$\mathbb{E}[\mathrm{Reg}(T)] \leq \frac{\lambda}{\eta} \mathrm{dc}\left(\mathcal{F}, M, T, \frac{\eta}{4\lambda}\right) + \frac{2T}{\lambda}\kappa(b/T^\beta) + \frac{6HT^{2-\beta}}{\lambda} + bT^{1-\beta},$$

*where the expectation is over the samples drawn from the MDP and the algorithm's internal randomness. Let $\mathrm{dc}(\mathcal{F}, M, T)$ be any bound on $\sup_{\mu \leq 1} \mathrm{dc}(\mathcal{F}, MT, \mu)$ and set $\eta = 1/4b^2$ and $\lambda = \sqrt{\frac{T\kappa(b/T^2)}{b^2\mathrm{dc}(\mathcal{F},M,T)}}$. If $\lambda b^2 \geq 1$, then our bound becomes*

$$\mathbb{E}[\mathrm{Reg}(T)] = O\left(b\sqrt{\mathrm{dc}\left(\mathcal{F}, M, T\right)\kappa(b/T^2)T} + \mathrm{dc}\left(\mathcal{F}, M, T\right) + \sqrt{H}\right). \qquad (5)$$

For the simpler form of our regret bound in Equation (5), we chose a specific $\lambda$ (the condition $\lambda b^2 \geq 1$ is easy to satisfy for large $T$). However, we may also use that $\lambda^2 \mathrm{dc}(\mathcal{F}, M, T, \eta/(4\lambda))$ is an increasing function of $\lambda$ to set $\lambda$ differently and achieve a better tuned bound. To instantiate the general regret bound in Theorem 1 to specific settings, we first derive bounds on the decoupling coefficient in Section 4.1 and then state and discuss Theorem 1 for those settings in Section 4.2.

## 4.1 Decoupling Coefficient and its Relation to Other Complexity Measures

We present several previously studied settings for which the decoupling coefficient is provably small.

**Linear Markov decision processes.** We first consider the linear MDP setting [Bradtke and Barto, 1996, Melo and Ribeiro, 2007] which was formally defined by Jin et al. [2020] as:

**Definition 3** (Linear MDP ). *An MDP with feature map $\phi : \mathcal{X} \times \mathcal{A} \to \mathbb{R}^d$ is linear, if for any $h \in [H]$, there exist $d$ unknown (signed) measures $\mu_h = (\mu_h^{(1)}, \ldots, \mu_h^{(d)})$ over $\mathcal{X}$ and an unknown vector $\theta_h \in \mathbb{R}^d$, such that for any $(x, a) \in \mathcal{X} \times \mathcal{A}$, we have*

$$P^h(\cdot \,|\, x, a) = \langle \phi(x,a), \mu_h(\cdot) \rangle, \quad r^h(x, a) = \langle \phi(x, a), \theta_h \rangle.$$

*We assume further $\|\phi(x, a)\| \leq 1$ for all $(x, a) \in \mathcal{X} \times \mathcal{A}$, and $\max\{\|\mu_h(\mathcal{X})\|, \|\theta_h\|\} \leq \sqrt{d}$ for all $h \in [H]$.*

Since the transition kernel and expected immediate rewards are linear in given features $\phi$, it is well known that the Q-function of any policy is also a linear function in $\phi$ [Jin et al., 2020]. We further show in the following proposition that the decoupling coefficient is also bounded by $O(dH \ln T)$:

**Proposition 1.** *In linear MDPs, the linear function class $\mathcal{F} = \bigotimes_{h=1}^H \{\langle \phi(\cdot, \cdot), f \rangle \,|\, f \in \mathbb{R}^d, \|f\| \leq (H + 1 - h)\sqrt{d}\}$ satisfies Assumptions 1-3, and the decoupling coefficient for $\mu \leq 1$ is bounded by*

$$\mathrm{dc}(\mathcal{F}, M, T, \mu) \leq 2dH(1 + \ln(2HT)).$$

Notably, since tabular MDPs are linear MDPs with dimension at most $|\mathcal{X}||\mathcal{A}|$, Proposition 1 implies that $\mathrm{dc}(\mathcal{F}, M, T, \mu) \leq 2|\mathcal{X}||\mathcal{A}|H(1 + \ln(2HT))$ in tabular MDPs. As compared to other complexity measures such as Eluder dimension, the bound of the decoupling coefficient generally exhibits an additional factor of $H$. This factor appears because the decoupling coefficient is defined for the sum of all time steps $[H]$ instead of the maximum. We chose the formulation with sums because it is advantageous when the complexity of the MDP of function class varies with $h$.

**Generalized linear MDPs.** Linear functions admit a straightforward generalization to include a rich class of non-linear functions which have previously been studied by Wang et al. [2019, 2020] in the RL setting.

**Definition 4** (Generalized linear models). *For a given link function $\sigma : [-1, 1] \to [-1, 1]$ such that $|\sigma'(x)| \in [k, K]$ for Lipschitz constants $0 < k \leq K < \infty$, and a known feature map $\phi : \mathcal{X} \times \mathcal{A} \to \mathbb{R}^d$, the class of generalized linear models is*

$$\mathcal{G} := \{(x, a) \to \sigma(\langle \phi(x, a), \theta \rangle \,|\, \theta \in \Theta \subset \mathbb{R}^d\}.$$

As the following result shows, we can readily bound the decoupling coefficient for generalized linear Q-functions.

**Proposition 2.** *If $(\mathcal{F}_h)_{h=1}^H$ are generalized linear models with Lipschitz constants $k, K$, and bounded norm $\|f\| \leq \sqrt{d}H$ for all $h$ and all $f \in \mathcal{F}_h$, then the decoupling coefficient for any $\mu \leq 1$ is bounded by*

$$\mathrm{dc}(\mathcal{F}, M, T, \mu) \leq 2dH \frac{K^2}{k^2}(1 + \ln(2HT)).$$

**Bellman-Eluder Dimension.** Finally in more generality, our decoupling coefficient is small for instances with low Bellman-Eluder dimension [Jin et al., 2021]. Problems with low Bellman-Eluder dimension include in decreasing order of generality: small Eluder dimension [Wang et al., 2020], generalized linear MDPs [Wang et al., 2019], linear MDPs [Bradtke and Barto, 1996, Melo and Ribeiro, 2007], and tabular MDPs. Before stating our reduction of Bellman Eluder dimension to decoupling coefficient formally, we first restate the definition of Bellman Eluder dimension in the following three definitions:

**Definition 5** ($\varepsilon$-independence between distributions). *Let $\mathcal{G}$ be a function class defined on $\mathcal{X}$, and $\nu, \mu_1, \ldots, \mu_n$ be probability measures over $\mathcal{X}$. We say $\nu$ is $\varepsilon$-independent of $\{\mu_1, \mu_2, \ldots, \mu_n\}$ with respect to $\mathcal{G}$ if there exists $g \in \mathcal{G}$ such that $\sqrt{\sum_{i=1}^n (\mathbb{E}_{\mu_i}[g])^2} \leq \varepsilon$, but $|\mathbb{E}_\nu[g]| > \varepsilon$.*

**Definition 6** ((Distributional Eluder (DE) dimension). *Let $\mathcal{G}$ be a function class defined on $\mathcal{X}$, and $\Pi$ be a family of probability measures over $\mathcal{X}$. The distributional Eluder dimension $\dim_{DE}(\mathcal{G}, \Pi, \varepsilon)$ is the length of the longest sequence $\{\rho_1, \ldots, \rho_n\} \subset \Pi$ such that there exists $\varepsilon' > \varepsilon$ where $\rho_i$ is $\varepsilon'$-independent of $\{\rho_1, \ldots, \rho_{i-1}\}$ for all $i \in [n]$.*

**Definition 7** (Bellman Eluder (BE) dimension [Jin et al., 2021]). *Let $\mathcal{E} := \bigotimes_{h=1}^H \{\mathcal{E}_h(f; x, a) : f \in \mathcal{F}\}$ be the set of Bellman residuals induced by $\mathcal{F}$ at step $h$, and $\Pi = \{\Pi_h\}_{h=1}^H$ be a collection of $H$ probability measure families over $\mathcal{X} \times \mathcal{A}$. The $\varepsilon$-Bellman Eluder dimension of $\mathcal{F}$ with respect to $\Pi$ is defined as*

$$\dim_{BE}(\mathcal{F}, \Pi, \varepsilon) := \max_{h \in [H]} \dim_{DE}(\mathcal{E}, \Pi, \varepsilon).$$

We show that the decoupling coefficient is small whenever the Bellman-Eluder dimension is small:

**Proposition 3.** *Let $\Pi = \mathcal{D}_\mathcal{F}$ be the set of probability measures over $\mathcal{X} \times \mathcal{A}$ at any step $h$ obtained by following the policy $\pi_f$ for $f \in \mathcal{F}$. If for all $\varepsilon > 0$*

$$\dim_{BE}(\mathcal{F}, \Pi, \varepsilon) \leq E \ln \frac{1}{\varepsilon},$$

*then the decoupling coefficient for any $\mu \leq 1$ is bounded by*
$$\mathrm{dc}(\mathcal{F}, M, T, \mu) \leq 4(1 + \ln(T))EH.$$

**Bellman-rank.** Another general complexity measure sufficient for provably efficient algorithms is the low Bellman-rank [Jiang et al., 2017], which includes reactive POMDPs. As discussed in Jin et al. [2021], a certain type of Bellman rank ("Q-type") implies bounded Bellman Eluder dimension (as defined above) and is thus also bounded decoupling coefficient. However, it is an open question if a low Bellman-rank in the original definition [Jiang et al., 2017] implies a small decoupling coefficient. The main difference is that Bellman-rank considers measures induced by $x \sim \pi_f, a \sim \pi_{f'}$, whereas the decoupling coefficient always samples state and action from the same policy.

### 4.2 Interpretation of Theorem 1

We can instantiate Theorem 1 with all bounds on the decoupling coefficients derived in the previous section. To illustrate the results, we will present two specific cases. The first one is for finite function classes that are often considered for simplicity [e.g. Jiang et al., 2017].

**Corollary 1** (Regret bound for finite function classes with completeness). *Assume a finite function class $\mathcal{F}$ that satisfies Assumptions 1, 2 and 3 with range $b = 2$. Assume further that the stagewise prior is uniform $p_0^h(f) = 1/|\mathcal{F}_h|$, and $|\mathcal{F}| = \prod_{h=1}^H |\mathcal{F}_h|$. Set parameters $\eta = 0.1$ and $\lambda = \sqrt{\frac{T \ln |\mathcal{F}|}{\mathrm{dc}(\mathcal{F}, M, T)}}$. Then the expected regret of Algorithm 1 after $T$ episodes is bounded on any MDP $M$ as*

$$\mathbb{E}[\mathrm{Reg}(T)] = O\left(\sqrt{\mathrm{dc}(\mathcal{F}, M, T)\, T \ln |\mathcal{F}|}\right).$$

Note that this result can be generalized readily to infinite function-classes when we replace $\ln |\mathcal{F}|$ by an appropriate covering number $\mathcal{N}_\infty(\mathcal{F}, \epsilon)$ for $\epsilon$ small enough.

In addition to finite function classes, we also illustrate Theorem 1 for linear Markov decision processes:

**Corollary 2.** *Assume Algorithm 1 is run on a $d$-dimensional linear MDP with the function class from Proposition 1. Assume further a learning rate of $\eta = 0.4H^{-2}$ and $\lambda = \sqrt{\frac{T\kappa(H/T^2)}{dH^3(1+\ln(2HT))}}$. Then the expected regret after $T$ episodes is bounded as*

$$\mathbb{E}[\mathrm{Reg}(T)] \leq O(H^{3/2}\sqrt{dT\kappa(H/T^2)\ln(HT)}).$$

*If the stage-wise priors $p_0^h$ are chosen uniformly, then $\kappa(\epsilon) = HdO(\ln(Hd\epsilon))$ as in (4), and thus*

$$\mathbb{E}[\mathrm{Reg}(T)] \leq O(H^2 d\sqrt{T}\ln(dHT)).$$

Our regret bound improves the regret bound of $\tilde{O}(H^{2.5}d^2\sqrt{T} + H^5 d^4)$ for the posterior-sampling method OPT-RLSVI [Zanette et al., 2020a] by a factor of $\sqrt{H}d$. It also improves the $\tilde{O}(d^{3/2}H^2\sqrt{T})$ regret of UCB-LSVI [Jin et al., 2020] by a factor of $\sqrt{d}$. However, we would like to remark that these algorithms are known to be computationally efficient in this setting, while the computational tractability of our method is an open problem. Our regret bound also matches the bound of Zanette et al. [2020b] without misspecification once we account for the different boundedness assumption ($b = 2$ instead of $b = H + 1$) in this work.

## 5 Proof Overview of Theorem 1

We provide a detailed proof of Theorem 1 in the appendix and highlight the main steps in this section. We start by presenting an alternate way to write the posterior in (3) that lends itself better to our analysis. To that end, we introduce some helpful notations.

We denote by $\zeta_s = \{x_s^h, a_s^h, r_s^h\}_{h\in[H]}$ the trajectory collected in the $s$-th episode. The notation $\mathbb{E}_{\pi_{f_s}}$ is equivalent to $\mathbb{E}_{\zeta_s \sim \pi_{f_s}}$. Moreover, in the following, the symbol $S_t$ at episode $t$ contains all historic observations up to episode $t$, which include both $\{\zeta_s\}_{s\in[t]}$ and $\{f_s\}_{s\in[t]}$. These observations are generated in the order $f_1 \sim p_0(\cdot), \zeta_1 \sim \pi_{f_1}, f_2 \sim p(\cdot|S_1), \zeta_2 \sim \pi_{f_2}, \ldots$.

We further define the TD error difference at episode $s$ as

$$\Delta L^h(f^h, f^{h+1}; \zeta_s) = (\quad f^h(x_s^h, a_s^h) - r_s^h - f^{h+1}(x_s^{h+1}))^2$$
$$-(\mathcal{T}_h^\star f^{h+1}(x_s^h, a_s^h) - r_s^h - f^{h+1}(x_s^{h+1}))^2.$$

The term we subtract from the TD error is the sampling error of this transition for the Bellman error. With $\Delta f^1(x^1) = f^1(x^1) - Q_1^\star(x^1)$ defined as the error of $f^1$ and

$$\hat{\Phi}_t^h(f) = -\ln p_0^h(f^h) + \eta \sum_{s=1}^{t-1} \Delta L^h(f^h, f^{h+1}; \zeta_s)$$

$$+ \ln \mathbb{E}_{\tilde{f}^h \sim p_0^h} \exp\left(-\eta \sum_{s=1}^{t-1} \Delta L^h(\tilde{f}^h, f^{h+1}; \zeta_s)\right),$$

we can equivalently rewrite the posterior distribution (3) in the following form:

$$p(f|S_{t-1}) \propto \exp\left(-\sum_{h=1}^H \hat{\Phi}_t^h(f) + \lambda \Delta f^1(x^1)\right).$$

Note that in the definition of $\hat{\Phi}$, we have replaced $L^h(\cdot)$ by $\Delta L^h(\cdot)$. Although we do not need to know the Bellman operator $\mathcal{T}_h^\star$ in the actual algorithm, in our theoretical analysis, it is equivalent to knowing the operator via the use of $\Delta L^h(\cdot)$. This equivalence is possible with the temporal difference error likelihood which we introduce in this paper, and this is the main technical reason why we choose this conditional posterior sampling distribution. It is the key observation that allows us to circumvent the double-sample issue discussed earlier.

With the new expression of posterior distribution, we can start the proof of Theorem 1 by using the following decomposition, referred to the value-function error decomposition in Jiang et al. [2017],

$$V_1^\star(x^1) - V_1^{\pi_{f_t}}(x^1) = \sum_{h=1}^{H} \mathbb{E}_{\pi_{f_t}}[\mathcal{E}_h(f_t, x_t^h, a_t^h)] - \Delta f_t^1(x^1).$$

We now write the expected instantaneous regret of episode $t$ (scaled by $\lambda$) using this decomposition as

$$\lambda \mathbb{E}_{S_{t-1}} \mathbb{E}_{f_t \sim p(\cdot|S_{t-1})}[V_1^\star(x^1) - V_1^{\pi_{f_t}}(x^1)]$$

$$= \underbrace{\mathbb{E}_{S_{t-1}} \mathbb{E}_{f_t \sim p(\cdot|S_{t-1})} \sum_{h=1}^{H} \left[ \lambda \mathbb{E}_{\pi_{f_t}} \mathcal{E}_h(f_t, x_t^h, a_t^h) - \frac{\eta}{4} \sum_{s=1}^{t-1} \mathbb{E}_{\pi_{f_s}} (\mathcal{E}_h(f_t; x_s^h, a_s^h))^2 \right]}_{F_t^{dc}}$$

$$+ \underbrace{\mathbb{E}_{S_{t-1}} \mathbb{E}_{f_t \sim p(\cdot|S_{t-1})} \left[ \sum_{h=1}^{H} \frac{\eta}{4} \sum_{s=1}^{t-1} \mathbb{E}_{\pi_{f_s}} (\mathcal{E}_h(f_t; x_s^h, a_s^h))^2 - \lambda \Delta f_t^1(x^1) \right]}_{F_t^\kappa}.$$

By summing over $t = 1, \ldots, T$, we obtain the following expression of cumulative expected regret

$$\lambda \mathbb{E} \operatorname{Reg}(T) = \lambda \sum_{t=1}^{T} \mathbb{E}_{S_{t-1}} \mathbb{E}_{f_t \sim p(\cdot|S_{t-1})}[V_1^\star(x^1) - V_1^{\pi_{f_t}}(x^1)] = \sum_{t=1}^{T} F_t^{dc} + \sum_{t=1}^{T} F_t^\kappa. \qquad (6)$$

To bound the first term on the RHS, we can use the definition of decoupling coefficient which gives that

$$\sum_{t=1}^{T} F_t^{dc} \leq \frac{\lambda^2}{\eta} dc\left(\mathcal{F}, M, T, \frac{\eta}{4\lambda}\right). \qquad (7)$$

To complete the proof is remains to upper-bound the $F_t^\kappa$ terms in (6). To do so we start with the following bounds (Lemma 1 and Lemma 4 in the appendix) which requires the realizability assumption:

$$\mathbb{E}_{f \sim p(\cdot|S_{t-1})}\left( \sum_{h=1}^{H} \hat{\Phi}_t^h(f) - \lambda \Delta f^1(x^1) + \ln p(f|S_{t-1}) \right)$$

$$= \inf_p \mathbb{E}_{f \sim p(\cdot)}\left( \sum_{h=1}^{H} \hat{\Phi}_t^h(f) - \lambda \Delta f^1(x^1) + \ln p(f) \right)$$

$$\leq \lambda \epsilon + 4\eta(t-1)H\epsilon^2 + \kappa(\epsilon). \qquad (8)$$

The occurrence of the term $\kappa(\epsilon)$ here implicitly corresponds to the realizability assumption. It can also be shown (see Lemma 10) that

$$\sum_{h=1}^{H} \mathbb{E}_{S_{t-1}} \mathbb{E}_{f \sim p(\cdot|S_{t-1})} \left[ \ln \mathbb{E}_{\tilde{f}^h \sim p_0^h} \exp\left( -\eta \sum_{s=1}^{t-1} \Delta L^h(\tilde{f}^h, f^{h+1}, \zeta_s) \right) \right]$$

$$\geq -\eta \epsilon(2b+\epsilon)(t-1)H - \kappa(\epsilon),$$

which means that the expected log-partition function in the definition of conditional posterior distribution is small, and its effect is under control. The proof requires the completeness assumption, and the occurrence of the term $\kappa(\epsilon)$ here implicitly corresponds to the completeness assumption. By combining this inequality with (8), and by using the definition of $\hat{\Phi}$, we obtain the following inequality

$$\mathbb{E}_{S_{t-1}} \mathbb{E}_{f \sim p(\cdot|S_{t-1})} \left( \eta \sum_{h=1}^{H} \sum_{s=1}^{t-1} \Delta L^h(f^h, f^{h+1}, \zeta_s) - \lambda \Delta f^1(x^1) + \ln \frac{p(f|S_{t-1})}{\prod_{h=1}^{H} p_0^h(f^h)} \right)$$

$$\leq \lambda \epsilon + \eta(t-1)H\epsilon(5\epsilon + 2b) + 2\kappa(\epsilon). \qquad (9)$$

In this bound, we have removed the effect of log-partition function in the definition of the conditional posterior probability, and the inequality bounds the expected cumulative squared TD error.

One can further establish a connection between squared TD error and Bellman residual, by showing (Lemma 5 and Lemma 8) that

$$
\mathbb{E}_{S_{t-1}} \mathbb{E}_{f \sim p(\cdot|S_{t-1})} \left[ \eta \sum_{h=1}^{H} \sum_{s=1}^{t-1} \Delta L^h(f^h, f^{h+1}, \zeta_s) + \ln \frac{p(f|S_{t-1})}{\prod_{h=1}^{H} p_0^h(f^h)} \right]
$$

$$
\geq \sum_{h=1}^{H} \mathbb{E}_{S_{t-1}} \mathbb{E}_{f \sim p(\cdot|S_{t-1})} \left[ \eta \sum_{s=1}^{t-1} \Delta L^h(f^h, f^{h+1}, \zeta_s) + 0.5 \ln \frac{p(f^h, f^{h+1}|S_{t-1})}{p_0^h(f^h) p_0^{h+1}(f^{h+1})} \right]
$$

$$
\geq 0.25 \eta \sum_{s=1}^{t-1} \sum_{h=1}^{H} \mathbb{E}_{S_{t-1}} \mathbb{E}_{f_t \sim p(\cdot|S_{t-1})} \mathbb{E}_{\pi_{f_s}} (\mathcal{E}_h(f; x_s^h, a_s^h))^2.
$$

By combining this bound with (9), we obtain

$$
F_t^\kappa = 0.25 \eta \sum_{s=1}^{t-1} \sum_{h=1}^{H} \mathbb{E}_{S_{t-1}} \mathbb{E}_{f_t \sim p(\cdot|S_{t-1})} \mathbb{E}_{\pi_{f_s}} (\mathcal{E}_h(f_t; x_s^h, a_s^h))^2 - \lambda \mathbb{E}_{S_{t-1}} \mathbb{E}_{f_t \sim p(\cdot|S_{t-1})} \Delta f_t^1(x^1)
$$

$$
\leq \lambda \epsilon + \eta(t-1) H \epsilon(5\epsilon + 2b) + 2\kappa(\epsilon).
$$

This implies the following bound for the sum of $F_t^\kappa$ term in (6):

$$
\sum_{t=1}^{T} F_t^\kappa \leq \lambda \epsilon T + \eta(t-1) H \epsilon(5\epsilon + 2b) T + 2\kappa(\epsilon) T.
$$

By combining this estimate with (7) and (6), we obtain

$$
\lambda \mathbb{E} \operatorname{Reg}(T) \leq \frac{\lambda^2}{\eta} \operatorname{dc}\left( \mathcal{F}, M, T, \frac{\eta}{4\lambda} \right) + \lambda \epsilon T + \eta(t-1) H \epsilon(5\epsilon + 2b) T + 2\kappa(\epsilon) T.
$$

The choice of $\epsilon = b/T^\beta$ implies the first bound of Theorem 1.

## 6  Conclusion

This paper proposed a new posterior sampling algorithm for episodic reinforcement learning using conditional sampling with a temporal difference error likelihood. We show that posterior sampling methods can achieve the same frequentist regret guarantees as algorithms based on optimism in the face of uncertainty (OFU) in a wide range of settings with general value function approximation. Our results thus suggest that there is no statistical efficiency gap between OFU and posterior sampling algorithms. While our results are stated in expectation, it is possible to derive high probability bounds with a slightly more complicated analysis.

One of the key open questions for provably efficient reinforcement learning under general assumptions such as low Bellman-Eluder dimension or Bellman rank is that of computational efficiency. No computationally tractable algorithm is known for such general settings. Although the computational complexity of sampling from the posterior in our algorithm is unknown, we hope that the addition of a sampling-based algorithm to the pool of sample-efficient algorithms in this setting may provide additional tools to design statistically and computationally tractable algorithms.

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
