# Appendix

## Contents of Main Article and Appendix

# A  Full Proof of Theorem 1

## A.1  Generalization of Theorem 1

Instead of proving Theorem 1 directly, we will prove a slightly more general version that we will state formally in Theorem 2. In short, this version considers a more general family of posteriors that include an extra parameter $\alpha \in (0, 1]$. The original posterior of Algorithm 1 in Theorem 1 corresponds to the case of $\alpha = 1$.

First, we introduce some notations used in the proof. We define
$$\text{Reg}(f) = (V_1^\star(x^1) - V_1^{\pi_f}(x^1)).$$
Given state action pair $[x^h, a^h]$, we use the notation $[x^{h+1}, r^h] \sim P^h(\cdot|x^h, a^h)$ to denote the joint probability of sampling the next state $x^{h+1} \sim P^h(\cdot|x^h, a^h)$ and reward $r^h \sim R^h(\cdot|x^h, a^h)$.

Let $\zeta_s = \{[x_s^h, a_s^h, r_s^h]\}_{h \in [H]}$ be the trajectory of the $s$-th episode. In the following, the notation $S_t$ at time $t$ includes all historic observations up to time $t$, which include both $\{\zeta_s\}_{s \in [t]}$ and $\{f_s\}_{s \in [t]}$. These observations are generated in the order $f_1 \sim p_0(\cdot)$, $\zeta_1 \sim \pi_{f_1}$, $f_2 \sim p(\cdot|S_1)$, $\zeta_2 \sim \pi_{f_2}$, ....

Define the excess loss
$$\Delta L^h(f^h, f^{h+1}; \zeta_s) = (f^h(x_s^h, a_s^h) - r_s^h - f^{h+1}(x_s^{h+1}))^2 \\ - (\mathcal{T}_h^\star f^{h+1}(x_s^h, a_s^h) - r_s^h - f^{h+1}(x_s^{h+1}))^2,$$
and define the potential $\hat{\Phi}$, which contains the extra parameter $\alpha$:
$$\hat{\Phi}_t^h(f) = -\ln p_0^h(f^h) + \alpha\eta \sum_{s=1}^{t-1} \Delta L^h(f^h, f^{h+1}; \zeta_s) \tag{10}$$
$$+ \alpha \ln \mathbb{E}_{\tilde{f}^h \sim p_0^h} \exp\left(-\eta \sum_{s=1}^{t-1} \Delta L^h(\tilde{f}^h, f^{h+1}; \zeta_s)\right),$$
and define
$$\Delta f^1(x^1) = f^1(x^1) - Q_1^\star(x^1),$$
where $Q_1^\star(x^1) = V_1^\star(x^1)$ using our notation. Given $S_{t-1}$, we may define the following generalized posterior probability $\hat{p}_t$ on $\mathcal{F}$:
$$\hat{p}_t(f) \propto \exp\left(-\sum_{h=1}^{H} \hat{\Phi}_t^h(f) + \lambda\Delta f^1(x^1)\right). \tag{11}$$

We will also introduce the following definition.

**Definition 8.** *We define for $\alpha \in (0, 1)$, and $\epsilon > 0$:*
$$\kappa^h(\alpha, \epsilon) = (1 - \alpha) \ln \mathbb{E}_{f^{h+1} \sim p_0^{h+1}} p_0^h\left(\mathcal{F}_h(\epsilon, f^{h+1})\right)^{-\alpha/(1-\alpha)},$$
*and we define $\kappa^h(1, \epsilon) = \lim_{\alpha \to 1^-} \kappa^h(\alpha, \epsilon)$.*

It is easy to check when $\alpha = 1$, the posterior distribution of (11) is equivalent to the posterior $p(f|S_{t-1})$ defined in (3).

When $\alpha = 1$,
$$\kappa^h(1, \epsilon) = \sup_{f^{h+1} \in \mathcal{F}_{h+1}} \ln \frac{1}{p_0^h\left(\mathcal{F}_h(\epsilon, f^{h+1})\right)} < \infty.$$

Therefore $\kappa(\epsilon)$ defined in Definition 1 can be written as
$$\kappa(\epsilon) = \sum_{h=1}^{H} \kappa^h(1, \epsilon).$$

However, the advantage of using a value $\alpha < 1$ is that $\kappa(\alpha, \epsilon)$ can be much smaller than $\kappa(1, \epsilon)$.

We will prove the following theorem for $\alpha \in (0, 1)$, which becomes Theorem 1 when $\alpha \to 1$.

**Theorem 2.** *Consider Algorithm 1 with the posterior sampling probability* (3) *replaced by* (11). *When $\eta b^2 \leq 0.4$, we have*

$$\sum_{t=1}^{T} \mathbb{E}_{S_{t-1}} \mathbb{E}_{f_t \sim \hat{p}_t} \operatorname{Reg}(f_t)$$

$$\leq \frac{\lambda}{\alpha \eta} \operatorname{dc}(\mathcal{F}, M, T, 0.25\alpha\eta/\lambda) + (T/\lambda) \sum_{h=1}^{H} \left[ \kappa^h(\alpha, \epsilon) - \ln p_0^h \Big( \mathcal{F}_h(\epsilon, Q_{h+1}^\star) \Big) \right]$$

$$+ \frac{\alpha}{\lambda} \eta \epsilon (5\epsilon + 2b) T(T-1)H + T\epsilon.$$

## A.2 Proof of Theorem 2

We need a number of technical lemmas. We start with the following inequality, which is the basis of our analysis.

**Lemma 1.**

$$\mathbb{E}_{f \sim \hat{p}_t} \left( \sum_{h=1}^{H} \hat{\Phi}_t^h(f) - \lambda \Delta f^1(x^1) + \ln \hat{p}_t(f) \right) = \inf_p \mathbb{E}_{f \sim p(\cdot)} \left( \sum_{h=1}^{H} \hat{\Phi}_t^h(f) - \lambda \Delta f^1(x^1) + \ln p(f) \right).$$

*Proof.* This is a direct consequence of the well-known fact that (11) is the minimizer of the right hand side. This fact is equivalent to the fact that the KL-divergence of any $p(\cdot)$ and $\hat{p}_t$ is non-negative. $\square$

We also have the following bound, which is needed to estimate the left hand side and right hand side of Lemma 1.

**Lemma 2.** *For all function $f \in \mathcal{F}$, we have*

$$\mathbb{E}_{[x_s^{h+1}, r_s^h] \sim P^h(\cdot|x_s^h, a_s^h)} \Delta L^h(f^h, f^{h+1}, \zeta_s) = (\mathcal{E}_h(f; x_s^h, a_s^h))^2.$$

*Moreover, we have*

$$\mathbb{E}_{[x_s^{h+1}, r_s^h] \sim P^h(\cdot|x_s^h, a_s^h)} \Delta L^h(f^h, f^{h+1}, \zeta_s)^2 \leq \frac{4b^2}{3} (\mathcal{E}_h(f; x_s^h, a_s^h))^2.$$

*Proof.* For notation simplicity, we introduce the random variable

$$Z = f^h(x_s^h, a_s^h) - r_s^h - f^{h+1}(x_s^{h+1}),$$

which depends on $[x_s^{h+1}, r_s^h]$, conditioned on $[x_s^h, a_s^h]$. The expecation $\mathbb{E}$ over $Z$ is with respect to the joint conditional probability $P^h(\cdot|x_s^h, a_s^h)$. Then

$$\mathbb{E}Z = \mathcal{E}_h(f; x_s^h, a_s^h),$$

and

$$\Delta L^h(f^h, f^{h+1}, \zeta_s) = Z^2 - (Z - \mathbb{E}Z)^2.$$

Since

$$\mathbb{E}[Z^2 - (Z - \mathbb{E}Z)^2] = (\mathbb{E}Z)^2,$$

we obtain

$$\mathbb{E}_{[x_s^{h+1}, r_s^h] \sim P^h(\cdot|x_s^h, a_s^h)} \Delta L^h(f^h, f^{h+1}, \zeta_s) = (\mathbb{E}Z)^2 = (\mathcal{E}_h(f; x_s^h, a_s^h))^2.$$

Also $Z \in [-b, b-1]$ and $\max Z - \min Z \leq b$ (when conditioned on $[x_s^h, a_s^h]$). This implies that

$$\mathbb{E}(Z^2 - (Z - \mathbb{E}Z)^2)^2 = (\mathbb{E}Z)^2[4\mathbb{E}Z^2 - 3(\mathbb{E}Z)^2] \leq \frac{4}{3}b^2(\mathbb{E}Z)^2.$$

We note that the maximum of $4\mathbb{E}Z^2 - 3(\mathbb{E}Z)^2$ is achieved with $Z \in \{-b, 0\}$ and $\mathbb{E}Z = -2b/3$. This leads to the second desired inequality. $\square$

The above lemma implies the following exponential moment estimate.

**Lemma 3.** *If $\eta b^2 \leq 0.8$, then for all function $f \in \mathcal{F}$, we have*

$$\ln \mathbb{E}_{[x_s^{h+1}, r_s^h] \sim P^h(\cdot | x_s^h, a_s^h)} \exp\Big( - \eta \Delta L^h(f^h, f^{h+1}, \zeta_s) \Big)$$

$$\leq \mathbb{E}_{[x_s^{h+1}, r_s^h] \sim P^h(\cdot | x_s^h, a_s^h)} \exp\Big( - \eta \Delta L^h(f^h, f^{h+1}, \zeta_s) \Big) - 1$$

$$\leq - 0.25\eta(\mathcal{E}_h(f; x_s^h, a_s^h))^2.$$

*Proof.* From $\eta b^2 \leq 0.8$, we know that

$$-\eta \Delta L^h(f^h, f^{h+1}, \zeta_s) \leq 0.8.$$

This implies that

$$\exp\Big( - \eta \Delta L^h(f^h, f^{h+1}, \zeta_s) \Big)$$

$$= 1 - \eta \Delta L^h(f^h, f^{h+1}, \zeta_s) + \eta^2 \psi\Big( - \eta \Delta L^h(f^h, f^{h+1}, \zeta_s) \Big) \Delta L^h(f^h, f^{h+1}, \zeta_s)^2$$

$$\leq 1 - \eta \Delta L^h(f^h, f^{h+1}, \zeta_s) + 0.67\eta^2 \Delta L^h(f^h, f^{h+1}, \zeta_s)^2$$

where we have used the fact that $\psi(z) = (e^z - 1 - z)/z^2$ is an increasing function of $z$, and $\psi(0.8) < 0.67$. It follows from Lemma 2 that

$$\ln \mathbb{E}_{[x_s^{h+1}, r_s^h] \sim P^h(\cdot | x_s^h, a_s^h)} \exp\Big( - \eta \Delta L^h(f^h, f^{h+1}, \zeta_s) \Big)$$

$$\leq \mathbb{E}_{[x_s^{h+1}, r_s^h] \sim P^h(\cdot | x_s^h, a_s^h)} \exp\Big( - \eta \Delta L^h(f^h, f^{h+1}, \zeta_s) \Big) - 1$$

$$\leq \mathbb{E}_{[x_s^{h+1}, r_s^h] \sim P^h(\cdot | x_s^h, a_s^h)} \Big[ -\eta \Delta L^h(f^h, f^{h+1}, \zeta_s) + 0.67\eta^2 \Delta L^h(f^h, f^{h+1}, \zeta_s)^2 \Big]$$

$$\leq - 0.25\eta(\mathcal{E}_h(f; x_s^h, a_s^h))^2,$$

where the first inequality is due to $\ln z \leq z - 1$. The last inequality used $0.67(4\eta b^2/3) < 0.75$ and Lemma 2. This proves the desired bound. $\qquad \square$

The following lemma upper bounds the right hand side of Lemma 1.

**Lemma 4.** *If $\eta b^2 \leq 0.8$, then*

$$\inf_p \ \mathbb{E}_{S_{t-1}} \mathbb{E}_{f \sim p(\cdot)} \Bigg[ \sum_{h=1}^H \hat{\Phi}_t^h(f) - \lambda \Delta f^1(x^1) + \ln p(f) \Bigg]$$

$$\leq \lambda \epsilon + 4\alpha\eta(t-1)H\epsilon^2 - \sum_{h=1}^H \ln p_0^h \big( \mathcal{F}_h(\epsilon, Q_{h+1}^\star) \big).$$

*Proof.* Consider any $f \in \mathcal{F}$. For any $\tilde{f}^h \in \mathcal{F}_h$ that only depends on $S_{s-1}$, we obtain from Lemma 3:

$$\mathbb{E}_{\zeta_s} \ \exp\Big( - \eta \Delta L^h(\tilde{f}^h, f^{h+1}, \zeta_s) \Big) - 1 \leq -0.25\eta \mathbb{E}_{\zeta_s} \ (\tilde{f}^h(x, a) - \mathcal{T}_h^\star f^{h+1}(x, a))^2 \leq 0. \quad (12)$$

Now, let

$$W_t^h = \mathbb{E}_{S_t} \mathbb{E}_{f \sim p(\cdot)} \ln \mathbb{E}_{\tilde{f}^h \sim p_0^h} \exp\Big( - \eta \sum_{s=1}^t \Delta L^h(\tilde{f}^h, f^{h+1}, \zeta_s) \Big),$$

then using the notation

$$\hat{q}_t^h(\tilde{f}^h | f^{h+1}, S_{t-1}) = \frac{\exp\Big( - \eta \sum_{s=1}^{t-1} \Delta L^h(\tilde{f}^h, f^{h+1}, \zeta_s) \Big)}{\mathbb{E}_{\tilde{f}'^h \sim p_0^h} \exp\Big( - \eta \sum_{s=1}^{t-1} \Delta L^h(\tilde{f}'^h, f^{h+1}, \zeta_s) \Big)},$$

we have

$$W_s^h - W_{s-1}^h = \mathbb{E}_{S_s} \mathbb{E}_{f \sim p(\cdot)} \ln \mathbb{E}_{\tilde{f}^h \sim \hat{q}_s^h(\cdot|f^{h+1}, S_{s-1})} \exp\left(-\eta \Delta L^h(\tilde{f}^h, f^{h+1}, \zeta_s)\right)$$

$$\leq \mathbb{E}_{S_s} \mathbb{E}_{f \sim p(\cdot)} \left(\mathbb{E}_{\tilde{f}^h \sim \hat{q}_s^h(\cdot|f^{h+1}, S_{s-1})} \exp\left(-\eta \Delta L^h(\tilde{f}^h, f^{h+1}, \zeta_s)\right) - 1\right) \leq 0,$$

where the first inequality is due to $\ln z \leq z - 1$, and the second inequality is from (12).

By noticing that $W_0^h = 0$, we obtain

$$W_t^h = W_0^h + \sum_{s=1}^{t} [W_s^h - W_{s-1}^h] \leq 0.$$

That is:

$$\mathbb{E}_{S_{t-1}} \mathbb{E}_{f \sim p(\cdot)} \ln \mathbb{E}_{\tilde{f}^h \sim p_0^h} \exp\left(-\eta \sum_{s=1}^{t-1} \Delta L^h(\tilde{f}^h, f^{h+1}, \zeta_s)\right) \leq 0. \qquad (13)$$

This implies that for an arbitrary $p(\cdot)$:

$$\mathbb{E}_{S_{t-1}} \mathbb{E}_{f \sim p(\cdot)} \left[\sum_{h=1}^{H} \hat{\Phi}_t^h(f) - \lambda \Delta f^1(x^1) + \ln p(f)\right]$$

$$= \mathbb{E}_{S_{t-1}} \mathbb{E}_{f \sim p(\cdot)} \left[ -\lambda \Delta f^1(x^1) + \alpha\eta \sum_{h=1}^{H} \sum_{s=1}^{t-1} \Delta L^h(f^h, f^{h+1}, \zeta_s) \right.$$

$$\left. + \alpha \sum_{h=1}^{H} \ln \mathbb{E}_{\tilde{f}^h \sim p_0^h} \exp\left(-\eta \sum_{s=1}^{t-1} \Delta L^h(\tilde{f}^h, f^{h+1}, \zeta_s)\right) + \ln \frac{p(f)}{p_0(f)}\right]$$

$$\leq \mathbb{E}_{S_{t-1}} \mathbb{E}_{f \sim p(\cdot)} \left[ -\lambda \Delta f^1(x^1) + \sum_{h=1}^{H} \alpha\eta \sum_{s=1}^{t-1} (\mathcal{E}_h(f; x_s^h, a_s^h))^2 + \ln \frac{p(f)}{p_0(f)}\right],$$

where in the derivation, the first equality used the definition of $\hat{\Phi}_t^h(f)$ in (10); the second inequality used (13), and then used the first equality of Lemma 2 to bound the expectation of $\Delta L(\cdot)$ by $\mathcal{E}_h$.

Note that if for all $h$

$$f^h \in \mathcal{F}_h(\epsilon, Q_{h+1}^\star),$$

then $|f(x_s^h, a_s^h) - Q_h^\star(x_s^h, a_s^h)| \leq \epsilon$. Therefore using the Bellman equation, we know

$$|\mathcal{E}_h(f; x_s^h, a_s^h)| \leq |f(x_s^h, a_s^h) - Q_h^\star(x_s^h, a_s^h)| + \sup |f(x_{s+1}^h) - Q_h^\star(x_{s+1}^h)| \leq 2\epsilon.$$

Therefore

$$\sum_{h=1}^{H} \alpha\eta \sum_{s=1}^{t-1} (\mathcal{E}_h(f; x_s^h, a_s^h))^2 \leq 4\alpha\eta H(t-1)\epsilon^2.$$

By taking $p(f) = p_0(f) I(f \in \mathcal{F}(\epsilon))/p_0(\mathcal{F}(\epsilon))$, with $\mathcal{F}(\epsilon) = \prod_h \mathcal{F}_h(\epsilon, Q_{h+1}^\star)$, we obtain the desired bound. □

The following lemma lower bounds the entropy term on the left hand side of Lemma 1.

**Lemma 5.** *We have*

$$\mathbb{E}_{f \sim \hat{p}_t(f)} \ln \hat{p}_t(f) \geq \alpha \mathbb{E}_{f \sim \hat{p}_t} \ln \hat{p}_t(f) + (1-\alpha) \mathbb{E}_{f \sim \hat{p}_t} \sum_{h=1}^{H} \ln \hat{p}_t(f^h)$$

$$\geq \frac{\alpha}{2} \sum_{h=1}^{H} \mathbb{E}_{f \sim \hat{p}_t} \ln \hat{p}_t(f^h, f^{h+1})$$

$$+ (1 - 0.5\alpha) \mathbb{E}_{f \sim \hat{p}_t} \ln \hat{p}_t(f^1) + (1-\alpha) \sum_{h=2}^{H} \mathbb{E}_{f \sim \hat{p}_t} \ln \hat{p}_t(f^h).$$

*Proof.* The first bound follows from the following inequality

$$\mathbb{E}_{f \sim \hat{p}_t} \ln \frac{\hat{p}_t(f)}{\prod_{h=1}^{H} \hat{p}_t(f^h)} \geq 0,$$

which is equivalent to the known fact that mutual information is non-negative (or KL-divergence is non-negative). The second inequality is equivalent to

$$\mathbb{E}_{f \sim \hat{p}_t} \ln \hat{p}_t(f) \geq 0.5 \mathbb{E}_{f \sim \hat{p}_t} \ln \hat{p}_t(f^1) + 0.5 \sum_{h=1}^{H} \mathbb{E}_{f \sim \hat{p}_t} \ln \hat{p}_t(f^h, f^{h+1}). \tag{14}$$

To prove (14), we consider the following two inequalities:

$$0.5 \mathbb{E}_{f \sim \hat{p}_t} \ln \hat{p}_t(f) \geq 0.5 \sum_{h=1}^{H} \mathbb{E}_{f \sim \hat{p}_t} \ln \hat{p}_t(f^h, f^{h+1}) I(h \text{ is a odd number})$$

and

$$0.5 \mathbb{E}_{f \sim \hat{p}_t} \ln \hat{p}_t(f) \geq 0.5 \mathbb{E}_{f \sim \hat{p}_t} \ln \hat{p}_t(f^1) + 0.5 \sum_{h=1}^{H} \mathbb{E}_{f \sim \hat{p}_t} \ln \hat{p}_t(f^h, f^{h+1}) I(h \text{ is an even number}).$$

Both follow from the fact that mutual information is non-negative. By adding the above two inequalities, we obtain (14). □

We will use the following decomposition to lower bound the left hand side of Lemma 1.

**Lemma 6.**

$$\mathbb{E}_{S_{t-1}} \mathbb{E}_{f \sim \hat{p}_t} \left( \sum_{h=1}^{H} \hat{\Phi}_t^h(f) - \lambda \Delta f^1(x^1) + \ln \hat{p}_t(f) \right)$$

$$\geq \underbrace{\mathbb{E}_{S_{t-1}} \mathbb{E}_{f \sim \hat{p}_t} \left[ -\lambda \Delta f^1(x^1) + (1 - 0.5\alpha) \ln \frac{\hat{p}_t(f^1)}{p_0^1(f^1)} \right]}_{A}$$

$$+ \sum_{h=1}^{H} 0.5\alpha \underbrace{\mathbb{E}_{S_{t-1}} \mathbb{E}_{f \sim \hat{p}_t} \left[ \eta \sum_{s=1}^{t-1} 2\Delta L^h(f^h, f^{h+1}, \zeta_s) + \ln \frac{\hat{p}_t(f^h, f^{h+1})}{p_0^h(f^h) p_0^{h+1}(f^{h+1})} \right]}_{B_h}$$

$$+ \sum_{h=1}^{H} \underbrace{\mathbb{E}_{S_{t-1}} \mathbb{E}_{f \sim \hat{p}_t} \left[ \alpha \ln \mathbb{E}_{\tilde{f}^h \sim p_0^h} \exp \left( -\eta \sum_{s=1}^{t-1} \Delta L^h(\tilde{f}^h, f^{h+1}, \zeta_s) \right) + (1 - \alpha) \ln \frac{\hat{p}_t(f^{h+1})}{p_0^{h+1}(f^{h+1})} \right]}_{C_h}.$$

*Proof.* We note from (10) that

$$\hat{\Phi}_t^h(f) = -\ln p_0^h(f^h) + \alpha\eta \sum_{s=1}^{t-1} \Delta L^h(f^h, f^{h+1}, \zeta_s)$$

$$+ \alpha \ln \mathbb{E}_{\tilde{f}^h \sim p_0^h} \exp \left( -\eta \sum_{s=1}^{t-1} \Delta L^h(\tilde{f}^h, f^{h+1}, \zeta_s) \right).$$

Now we can simply apply the second inequality of Lemma 5. □

We have the following result for $A$ in Lemma 6.

**Lemma 7.** *We have*

$$A \geq -\lambda \mathbb{E}_{S_{t-1}} \mathbb{E}_{f_t \sim \hat{p}_t(\cdot)} \Delta f_t^1(x^1).$$

*Proof.* This follows from the fact that the following KL-divergence is nonnegative:

$$\mathbb{E}_{f_t \sim \hat{p}_t} \ln \frac{\hat{p}_t(f_t^1)}{p_0^1(f_t^1)} \geq 0.$$

$\square$

The following proposition is from Zhang [2005] . The proof is included for completeness.

**Proposition 4.** *For each fixed $f \in \mathcal{F}$, we define a random variable for all $s$ and $h$ as follows:*

$$\xi_s^h(f^h, f^{h+1}, \zeta_s) = -2\eta\Delta L^h(f^h, f^{h+1}, \zeta_s)$$

$$- \ln \mathbb{E}_{[x_s^{h+1}, r_s^h] \sim P^h(\cdot|x_s^h, a_s^h)} \exp\left(-2\eta\Delta L^h(f^h, f^{h+1}, \zeta_s)\right).$$

*Then for all $h$:*

$$\mathbb{E}_{S_{t-1}} \exp\left(\sum_{s=1}^{t-1} \xi_s^h(f^h, f^{h+1}, \zeta_s)\right) = 1.$$

*Proof.* We can prove the proposition by induction. Assume that the equation

$$\mathbb{E}_{S_{t'-1}} \exp\left(\sum_{s=1}^{t'-1} \xi_s^h(f^h, f^{h+1}, \zeta_s)\right) = 1$$

holds for some $1 \leq t' < t$. Then

$$\mathbb{E}_{S_{t'}} \exp\left(\sum_{s=1}^{t'} \xi_s^h(f^h, f^{h+1}, \zeta_s)\right)$$

$$=\mathbb{E}_{S_{t'-1}} \exp\left(\sum_{s=1}^{t'-1} \xi_s^h(f^h, f^{h+1}, \zeta_s)\right) \mathbb{E}_{f_{t'} \sim p(\cdot|S_{t'-1})} \cdot \mathbb{E}_{\zeta_{t'} \sim \pi_{f_{t'}}} \exp\left(\xi_{t'}^h(f^h, f^{h+1}, \zeta_{t'})\right)$$

$$=\mathbb{E}_{S_{t'-1}} \exp\left(\sum_{s=1}^{t'-1} \xi_s^h(f^h, f^{h+1}, \zeta_s)\right) = 1.$$

Note that in the derivation, we have used the fact that

$$\mathbb{E}_{\zeta_{t'} \sim \pi_{f_{t'}}} \exp\left(\xi_{t'}^h(f^h, f^{h+1}, \zeta_{t'})\right) = 1.$$

The desired result now follows from induction. $\square$

The following lemma bounds $B_h$ in Lemma 6. This is a key estimate in our analysis.

**Lemma 8.** *Assume $\eta b^2 \leq 0.4$, then*

$$B_h \geq 0.25\alpha\eta \sum_{s=1}^{t-1} \mathbb{E}_{S_{t-1}} \mathbb{E}_{f \sim \hat{p}_t} \mathbb{E}_{\pi_{f_s}} (\mathcal{E}_h(f; x_s^h, a_s^h))^2.$$

*Proof.* Given any fixed $f \in \mathcal{F}$, we consider the random variable $\xi_s^h$ in Proposition 4. It follows that

$$\mathbb{E}_{f \sim \hat{p}_t} \left[\sum_{s=1}^{t-1} -\xi_s^h(f^h, f^{h+1}, \zeta_s) + \ln \frac{\hat{p}_t(f^h, f^{h+1})}{p_0^h(f^h)p_0^{h+1}(f^{h+1})}\right]$$

$$\geq \inf_p \mathbb{E}_{f \sim p} \left[\sum_{s=1}^{t-1} -\xi_s^h(f^h, f^{h+1}, \zeta_s) + \ln \frac{p(f^h, f^{h+1})}{p_0^h(f^h)p_0^{h+1}(f^{h+1})}\right]$$

$$= -\ln \mathbb{E}_{f^h \sim p_0^h} \mathbb{E}_{f^{h+1} \sim p_0^{h+1}} \exp\left(\sum_{s=1}^{t-1} \xi_s^h(f^h, f^{h+1}, \zeta_s)\right),$$

In the above derivation, the last equation used the fact that the minimum over $p$ is achieved at

$$p(f^h, f^{h+1}) \propto p_0^h(f^h) p_0^{h+1}(f^{h+1}) \exp \left( \sum_{s=1}^{t-1} \xi_s^h(f^h, f^{h+1}, \zeta_s) \right).$$

This implies that

$$\mathbb{E}_{S_{t-1}} \mathbb{E}_{f \sim \hat{p}_t} \left[ \sum_{s=1}^{t-1} -\xi_s^h(f^h, f^{h+1}, \zeta_s) + \ln \frac{\hat{p}_t(f^h, f^{h+1})}{p_0^h(f^h) p_0^{h+1}(f^{h+1})} \right]$$

$$\geq -\mathbb{E}_{S_{t-1}} \ln \mathbb{E}_{f^h \sim p_0^h} \mathbb{E}_{f^{h+1} \sim p_0^{h+1}} \exp \left( \sum_{s=1}^{t-1} \xi_s^h(f^h, f^{h+1}, \zeta_s) \right)$$

$$\geq -\ln \mathbb{E}_{f^h \sim p_0^h} \mathbb{E}_{f^{h+1} \sim p_0^{h+1}} \mathbb{E}_{S_{t-1}} \exp \left( \sum_{s=1}^{t-1} \xi_s^h(f^h, f^{h+1}, \zeta_s) \right) = 0.$$

The derivation used the concavity of $\log$ and Proposition 4. Now in the definition of $\xi_s^h(\cdot)$, We can use Lemma 3 to obtain the bound

$$\ln \mathbb{E}_{[x_s^{h+1}, r_s^h] \sim P^h(\cdot | x_s^h, a_s^h)} \exp \left( -2\eta \Delta L^h(f^h, f^{h+1}, \zeta_s) \right) \leq -0.5\eta(\mathcal{E}_h(f; x_s^h, a_s^h))^2,$$

which implies the desired result. $\qquad \square$

The following lemma bounds $C_h$ in Lemma 6.

**Lemma 9.** *We have for all $h \geq 1$:*

$$C_h \geq -(1-\alpha)]\mathbb{E}_{S_{t-1}} \ln \mathbb{E}_{f^{h+1} \sim p_0^{h+1}} \left( \mathbb{E}_{f^h \sim p_0^h} \exp \left( -\eta \sum_{s=1}^{t-1} \Delta L^h(f^h, f^{h+1}, \zeta_s) \right) \right)^{-\alpha/(1-\alpha)}.$$

*Proof.* We have

$$\mathbb{E}_{f \sim \hat{p}_t} \left[ \alpha \ln \mathbb{E}_{\tilde{f}^h \sim p_0^h} \exp \left( -\eta \sum_{s=1}^{t-1} \Delta L^h(\tilde{f}^h, f^{h+1}, \zeta_s) \right) + (1-\alpha) \ln \frac{\hat{p}_t(f^{h+1})}{p_0^{h+1}(f^{h+1})} \right]$$

$$\geq (1-\alpha) \inf_{p^h} \mathbb{E}_{f \sim p^h} \left[ \frac{\alpha}{1-\alpha} \ln \mathbb{E}_{\tilde{f}^h \sim p_0^h} \exp \left( -\eta \sum_{s=1}^{t-1} \Delta L^h(\tilde{f}^h, f^{h+1}, \zeta_s) \right) + \ln \frac{p^h(f^{h+1})}{p_0^{h+1}(f^{h+1})} \right]$$

$$= -(1-\alpha) \ln \mathbb{E}_{f^{h+1} \sim p_0^{h+1}} \left( \mathbb{E}_{f^h \sim p_0^h} \exp \left( -\eta \sum_{s=1}^{t-1} \Delta L^h(f^h, f^{h+1}, \zeta_s) \right) \right)^{-\alpha/(1-\alpha)},$$

where the inf over $p^h$ is achieved at

$$p^h(f^{h+1}) \propto p_0^{h+1}(f^{h+1}) \left( \mathbb{E}_{f^h \sim p_0^h} \exp \left( -\eta \sum_{s=1}^{t-1} \Delta L^h(f^h, f^{h+1}, \zeta_s) \right) \right)^{-\alpha/(1-\alpha)}.$$

This leads to the lemma. $\qquad \square$

The above bound implies the following estimate of $C_h$ in Lemma 6, which is easier to interpret.

**Lemma 10.** *For all $h \geq 1$,*

$$C_h \geq -\alpha\eta\epsilon(2b+\epsilon)(t-1) - \kappa^h(\alpha, \epsilon).$$

*Proof.* For $f^h \in \mathcal{F}_h(\epsilon, f^{h+1})$, we have

$$|\Delta L^h(f^h, f^{h+1}, \zeta_s)| \leq (\mathcal{E}_h(f, x_s^h, a_s^h))^2 + 2b|\mathcal{E}_h(f, x_s^h, a_s^h)| \leq \epsilon(2b+\epsilon).$$

It follows that

$$\mathbb{E}_{f^h \sim p_0^h} \exp \left( -\eta \sum_{s=1}^{t-1} \Delta L^h(f^h, f^{h+1}, \zeta_s) \right) \geq p_0^h(\mathcal{F}_h(\epsilon, f^{h+1})) \exp \left( -\eta(t-1)(2b+\epsilon)\epsilon \right).$$

This implies the bound. $\qquad \square$

The following result, referred to as the value-function error decomposition in Jiang et al. [2017], is well-known.

**Proposition 5** (Jiang et al. [2017]). *Given any $f_t$. Let $\zeta_t = \{[x_t^h, a_t^h, r_t^h]\}_{h \in [H]} \sim \pi_{f_t}$ be the trajectory of the greedy policy $\pi_{f_t}$, we have*

$$\text{Reg}(f_t) = \mathbb{E}_{\zeta_t \sim \pi_{f_t}} \sum_{h=1}^{H} \mathcal{E}_h(f_t, x_t^h, a_t^h) - \Delta f^1(x^1).$$

Equipped with all technical results above, we are ready to state the assemble all parts in the proof of Theorem 2:

*Proof of Theorem 2.* Let

$$\delta_t^h = \lambda \mathcal{E}_h(f_t, x_t^h, a_t^h) - 0.25\alpha\eta \sum_{s=1}^{t-1} \mathbb{E}_{\pi_s} \left( \mathcal{E}_h(f_t, x_s^h, a_s^h) \right)^2.$$

Then from the definition of decoupling coefficient, we obtain

$$\sum_{t=1}^{T} \mathbb{E}_{S_{t-1}} \mathbb{E}_{f_t \sim \hat{p}_t} \mathbb{E}_{\zeta_t \sim \pi_{f_t}} \sum_{h=1}^{H} \delta_t^h \leq \frac{\lambda^2}{\alpha\eta} \text{dc}(\mathcal{F}, M, T, 0.25\alpha\eta/\lambda). \tag{15}$$

From Proposition 5, we obtain

$$\mathbb{E}_{S_{t-1}} \mathbb{E}_{f_t \sim \hat{p}_t} \lambda \text{Reg}(f_t) - \mathbb{E}_{S_{t-1}} \mathbb{E}_{f_t \sim \hat{p}_t} \mathbb{E}_{\zeta_t \sim \pi_{f_t}} \sum_{h=1}^{H} \delta_t^h$$

$$= -\lambda \mathbb{E}_{S_{t-1}} \mathbb{E}_{f_t \sim \hat{p}_t} \Delta f_t^1(x_t^1) + 0.25\alpha\eta \sum_{h=1}^{H} \sum_{s=1}^{t-1} \mathbb{E}_{S_{t-1}} \mathbb{E}_{f_t \sim \hat{p}_t} \mathbb{E}_{\pi_{f_s}} \left( \mathcal{E}_h(f_t, x_s^h, a_s^h) \right)^2$$

$$\leq \mathbb{E}_{S_{t-1}} \mathbb{E}_{f \sim \hat{p}_t} \left( \sum_{h=1}^{H} \hat{\Phi}_t^h(f) - \lambda \Delta f^1(x^1) + \ln \hat{p}_t(f) \right) + \alpha\eta\epsilon(2b+\epsilon)(t-1)H + \sum_{h=1}^{H} \kappa^h(\alpha, \epsilon)$$

$$= \mathbb{E}_{S_{t-1}} \inf_p \mathbb{E}_{f \sim p} \left( \sum_{h=1}^{H} \hat{\Phi}_t^h(f) - \lambda \Delta f^1(x^1) + \ln p(f) \right)$$

$$+ \alpha\eta\epsilon(2b+\epsilon)(t-1)H + \sum_{h=1}^{H} \kappa^h(\alpha, \epsilon)$$

$$\leq \lambda\epsilon + \alpha\eta\epsilon(\epsilon + 4\epsilon + 2b)(t-1)H - \sum_{h=1}^{H} \ln p_0^h(\mathcal{F}(\epsilon, Q_{h+1}^\star)) + \sum_{h=1}^{H} \kappa^h(\alpha, \epsilon).$$

The first equality used the definition of $\delta_t^h$. The first inequality used Lemma 6 and Lemma 7 and Lemma 8 and Lemma 10. The second equality used Lemma 1. The second inequality follows from Lemma 4.

By summing over $t = 1$ to $t = T$, and use (15), we obtain the desired bound. $\square$

We are now ready to prove Theorem 1. Note that

$$-\ln p_0^h(\mathcal{F}(\epsilon, Q_{h+1}^\star)) \leq \kappa^h(1, \epsilon),$$

we have

$$-\sum_{h=1}^{H} \ln p_0^h(\mathcal{F}(\epsilon, Q_{h+1}^\star)) + \sum_{h=1}^{H} \kappa^h(\alpha, \epsilon) \leq 2\kappa(\epsilon).$$

By taking $\epsilon = b/T^\beta$, we obtain the desired result.

# B   Proofs for Decoupling Coefficient Bounds

## B.1   Proof of Proposition 1 (Linear MDP)

*Proof of Proposition 1.* Completeness follows from the fact that the $Q$ function of any policy $\pi$ is linear for linear MDPs. This follows directly from the Bellman equation.

$$Q_h^\pi(x,a) = r^h(x,a) + \mathbb{E}_{x' \sim P^h}[V_{h+1}^\pi(x')] = \langle \phi(x,a), \theta_h \rangle + \int_{\mathcal{S}} V_{h+1}^\pi(x') \langle \phi(x,a), d\,\mu_h(x') \rangle$$
$$= \langle \phi(x,a), w_h^\pi \rangle \,,$$

where $w_h^\pi = \theta_h + \int_{\mathcal{S}} V_{h+1}^\pi(x') d\,\mu_h(x')$. Hence the optimal $Q$-function is iin the function class.

Boundedness follows from $||\phi(x,a)|| \leq 1$ and $||f|| \leq (H+1)\sqrt{d}$.

Completeness follows by

$$[\mathcal{T}_h^\star f^{h+1}](x,a) = r^h(x,a) + \mathbb{E}_{x' \sim P^h}[\max_{a' \in \mathcal{A}} f^{h+1}(x',a')]$$
$$= \langle \phi(x,a), \theta_h \rangle + \int_{\mathcal{S}} \max_{a' \in \mathcal{A}} f^{h+1}(x',a') \langle \phi(x,a), d\,\mu_h(x') \rangle = \langle \phi(x,a), v_h^\pi \rangle \,,$$

where $v_h^\pi = \theta_h + \int_{\mathcal{S}} \max_{a' \in \mathcal{A}} f^{h+1}(x',a') d\,\mu_h(x')$.

**Bounding the decoupling coefficient.**    By the same argument, the Bellman error is linear
$$\mathcal{E}_h(f;x,a) = \langle \phi(x,a), w^h(f) \rangle$$
for some $w^h(f) \in \mathbb{R}^d, ||w^h(f)|| \leq \sqrt{d}H$. Denote $\phi_s^h = \mathbb{E}_{\pi_{f_s}}[\phi(x^h,a^h)]$ and $\Phi_t^h = \lambda I + \sum_{s=1}^{t} \phi(x^h,a^h)\phi(x^h,a^h)^\top$.

$$\mathbb{E}_{\pi_{f_t}}[\mathcal{E}_h(f_t; x_t^h, a_t^h)] - \mu \sum_{s=1}^{t-1} \mathbb{E}_{\pi_{f_s}}[\mathcal{E}_h(f_t; x_s^h, a_s^h)^2]$$

$$= w^h(f_t)^\top \phi_t^h - \mu w^h(f_t)^\top \sum_{s=1}^{t-1} \mathbb{E}_{\pi_{f_s}}[\phi(x^h,a^h)\phi(x^h,a^h)^\top] w^h(f_t)$$

$$\leq w^h(f_t)^\top \phi_t^h - \mu w^h(f_t)^\top \Phi_{t-1}^h w^h(f_t) + \mu\lambda dH^2$$

$$\leq \frac{1}{4\mu}(\phi_t^h)^\top (\Phi_{t-1}^h)^{-1}\phi_t^h + \mu\lambda dH^2 \,,$$

where the first inequality uses Jensen's inequality and the second is GM-AM inequality. Summing over all terms yields

$$\sum_{t=1}^{T} \sum_{h=1}^{H} \left[ \mathbb{E}_{\pi_{f_t}}[\mathcal{E}_h(f_t; x_t^h, a_t^h)] - \mu \sum_{s=1}^{t-1} \mathbb{E}_{\pi_{f_s}}[\mathcal{E}_h(f_t; x_s^h, a_s^h)^2] \right]$$

$$\leq \sum_{h=1}^{H} \left[ \frac{\ln(\det(\Phi_T^h)) - d\ln(\lambda)}{4\mu} + \lambda\mu C_1 T \right]$$

$$\leq H\left( \frac{d\ln(\lambda + T/d) - d\ln(\lambda)}{4\mu} + \lambda\mu dH^2 T \right).$$

Setting $\lambda = \min\{1, \frac{1}{4\mu^2 H^2 T}\}$ finishes the proof.   $\square$

## B.2   Proof of Proposition 2 (Generalized Linear Value Functions)

*Proof of Proposition 2.* We assume w.l.o.g. that $k \leq 1 \leq K$, otherwise we can scale the features and the link function accordingly. By completeness assumption, there exists a $g_t^h \in \mathcal{F}_h$, such that $g_t^h = \mathcal{T}_h^\star(f_t^{h+1})$. The Bellman error is

$$\mathcal{E}_h(f;x,a) = \sigma(\langle \phi(s,a), f_t^h \rangle - \mathcal{E}_h(f;x,a) = \sigma(\langle \phi(s,a), g_t^h \rangle \,.$$

By the Lipschitz property, we have for all $s \in [t]$

$$k|\langle \phi(x,a), w(f_s) \rangle| \le |\mathcal{E}_h(f_s; x, a)| \le K|\langle \phi(x,a), w^h(f_s) \rangle|$$

for $w^h(f_s) = f_s^h - g_s^h \in \mathbb{R}^d$.

The remaining proof is analogous to the previous one. Denote $\phi_s^h = \mathbb{E}_{\pi_{f_s}}[\phi(x^h, a^h)]$ and $\Phi_t^h = \lambda I + \sum_{s=1}^{t} \phi(x^h, a^h)\phi(x^h, a^h)^\top$.

$$\mathbb{E}_{\pi_{f_t}}[\mathcal{E}_h(f_t; x_t^h, a_t^h)] - \mu \sum_{s=1}^{t-1} \mathbb{E}_{\pi_{f_s}}[\mathcal{E}_h(f_t; x_s^h, a_s^h)^2]$$

$$\le K|w^h(f_t)^\top \phi_t^h| - \mu k^2 w^h(f_t)^\top \sum_{s=1}^{t-1} \mathbb{E}_{\pi_{f_s}}[\phi(x^h, a^h)\phi(x^h, a^h)^\top]w^h(f_t)$$

$$\le K|w^h(f_t)^\top \phi_t^h| - \mu k^2 w^h(f_t)^\top \Phi_{t-1}^h w^h(f_t) + \lambda \mu k^2 dH^2$$

$$\le \frac{K^2}{4\mu k^2}(\phi_t^h)^\top (\Phi_{t-1}^h)^{-1}\phi_t^h + \mu k^2 \lambda dH^2 \,,$$

where the first inequality uses Jensen's inequality and the second is GM-AM inequality. Summing over all terms yields

$$\sum_{t=1}^{T}\sum_{h=1}^{H}\left[\mathbb{E}_{\pi_{f_t}}[\mathcal{E}_h(f_t; x_t^h, a_t^h)] - \mu \sum_{s=1}^{t-1}\mathbb{E}_{\pi_{f_s}}[\mathcal{E}_h(f_t; x_s^h, a_s^h)^2]\right]$$

$$\le \sum_{h=1}^{H}K^2\left[\frac{\ln(\det(\Phi_T^h)) - d\ln(\lambda)}{4\mu k^2} + \lambda \mu k^2 C_1 T\right]$$

$$\le HK^2\left(\frac{d\ln(\lambda + T/d) - d\ln(\lambda)}{4\mu k^2} + \lambda \mu k^2 dH^2 T\right).$$

Setting $\lambda = \min\{1, \frac{1}{4\mu^2 k^2 H^2 T}\}$ finishes the proof. $\qquad\square$

## B.3 Proof of Proposition 3 (Bellman-Eluder dimension Reduction)

We require the following Lemma to prove the reduction of Bellman-Eluder dimension to the decoupling coefficient.

**Lemma 11.** *Let $\mu_1, \mu_2, \ldots \mu_{t-1}$ denote the measures over $S \times A$ obtained by following the policy induced by $(f_s)_{s=1}^{t-1}$ at stage $h$ and $\{\nu_1, \ldots, \nu_M\}$ be the set of unique measures in this set in decreasing order of occurrences and let $(N_i)_{i=1}^{M}$ be the number of times a measure appears in the sequence. If the the $\varepsilon$-Belmman-Eluder Dimension is $E$ and $|\mathbb{E}_{x,a\sim\mu_s}[\mathcal{E}_h(f_t; x, a)]| > \varepsilon$, then*

$$\sum_{s=1}^{t-1}\mathbb{E}_{x,a\sim\mu_s}[\mathcal{E}_h(f_t; x, a)^2] \ge w_t^h(\mathbb{E}_{x,a\sim\mu_t}[\mathcal{E}_h(f_t; x, a)])^2$$

$$\text{where } w_t^h = \begin{cases} N_i & \text{if } \mu_t = \nu_i \wedge i \in [E-1] \\ \lceil \frac{\sum_{i=E}^{M}N_i}{E} \rceil & \text{otherwise.} \end{cases}$$

*Proof.* If $\mu_t = \nu_i$, then the statement follows from Jensen's inequality. Otherwise by by the Bellman-Eluder dimension, for any set $(\mu_i')_{i=1}^{E}$ of pairwise different measures, it holds that

$$\sum_{i=1}^{E}\mathbb{E}_{x,a\sim\mu_i'}[\mathcal{E}_h(f_t; x, a)^2] \ge (\mathbb{E}_{x,a\sim\mu_t}[\mathcal{E}_h(f_t; x, a)])^2 \,.$$

It remains to show that we can construct at least $\lceil \frac{\sum_{i=E}^{M}N_i}{E} \rceil$ sets of $E$ pairwise different measures. This follows trivially by selecting sets greedily from the largest remaining duplicates of measures. $\quad\square$

Equipped with this lemma, we can now present the proof of Proposition 3:

*Proof of Proposition 3.* Denote $\epsilon_{t,s}^h = \mathbb{E}_{[x_s^h, a_s^h]}[\mathcal{E}_h(f_t; x, a)]$, the LHS is

$$\sum_{t=1}^T \sum_{h=1}^H \epsilon_{tt}^h \leq EH + \epsilon TH + \sum_{t=E+1}^T \sum_{h=1}^H \epsilon_{tt}^h \mathbb{I}\{\epsilon_{tt}^h > \epsilon\}.$$

For any $h \in [H]$, the RHS is bounded by Jensen's inequality, AM-GM inequality and Cauchy-Schwarz

$$\mu \sum_{t=1}^T \sum_{s=1}^{t-1} \epsilon_{ts}^{h\,2} + \frac{2E(1+\ln(T))}{4\mu} \geq \sqrt{2E(1+\ln(T)) \sum_{t=E+1}^T w_t^h \epsilon_{tt}^{h\,2} \mathbb{I}\{\epsilon_{tt}^h > \epsilon\}}$$

$$\geq \sqrt{\frac{2E(1+\ln(T))}{\sum_{t=E+1}^T \frac{1}{w_t^h}}} \sum_{t=E+1}^T \epsilon_{tt}^h \mathbb{I}\{\epsilon_{tt}^h > \epsilon\}.$$

Finally we need to bound the sum of weights $\sum_{t=1}^T \frac{1}{w_t^h}$, which are defined in Lemma 11. Every time the measure $\mu_t$ is in the set of the $E-1$ most common measures, one of the counts $N_i$ for $i \in [E-1]$ increases. Otherwise the count $\sum_{i \geq E} N_i$ increases by 1. Hence

$$\sum_{t=1}^T \frac{1}{w_t^h} \leq \sum_{i=1}^{E-1} \sum_{t=1}^T \frac{1}{t} + \sum_{t=1}^T \frac{E}{t} \leq 2E(1+\ln(T)).$$

$\square$