# OpenReview forum: "A Provably Efficient Model-Free Posterior Sampling Method for Episodic Reinforcement Learning"
_NeurIPS.cc/2021/Conference — NeurIPS 2021 Poster_

### Official Review · Reviewer_f4DA · 2021-07-15

**Rating:** 7
**Confidence:** 4

**Summary:**

In this paper, the authors study the model-free posterior sampling algorithm for the episodic RL problem. A novel algorithm and analysis are proposed. The regret guarantee is proved, and specifically it depends on a term related to the function class, a term related to the structural complexity measure (decoupling coefficient), and other parameters. The authors also show the decoupling coefficient is small for linear MDPs, generalized linear MDPs, and Bellman-Eluder Dimension. When specialized to linear MDP, the regret bound matches Zanette et al. [2020b], thus statistically efficient. However, the algorithm suffers from computationally inefficiency.


-----
I have read the response and other reviews. I'll keep my original rating.

**Main Review:**

I really enjoy reading the paper and would like to thank the authors. As we know that most TS related algorithms are restricted to the linear MDP setting, this paper makes significant contribution by extending the framework to the more general settings. The definition of the prior and the likelihood function, and the analysis are new to me. Though the regret bound in Theorem 1 is hard to parse, the authors then instantiate that to the linear MDP setting and compare them with the existing results. Overall, the main text is easy to follow. The technical part is dense. I don’t have time to check the proof, so I am not able to evaluate the correctness of the result.

-	Recently, we have seen a boom of paper beyond the linear MDP setting in theoretical RL research. However, most algorithms are not related to TS. The authors propose a new algorithm that can handle the more general setting. They also introduce new quantities about the function class and the structural complexity measures and bring new analysis to the community. I think this initial step may boost the research in the related subarea.
-	I notice that the optimistic prior is not symmetric for all level h since it only includes $\exp(\lambda f^1(x^1))$. Can you comment on why this special definition is helpful for the analysis?
-	Theorem 1 itself is hard to parse, since we generally don’t know how large the $\kappa$ terms and the decompiling coefficient terms are. I think the magnitude of $\kappa$ is only investigated for the linear MDP case. Is it possible to obtain similar bounds of $\kappa$ for other settings?
-	One limitation of the algorithm is the computationally inefficiency. Besides, I think the algorithm requires the know ledge of $\kappa$ and decompiling coefficient terms. Is this true? If so, is it possible to extend to the unknown case?
-	In Corollary 2, it seems that the priors are assumed to chosen uniformly so that a bound on $\kappa$ can be achieved. Is there any difficulty to extend it to more general priors?
-	The regret bounds are the expected regret bounds. It is generally believed that the high-probability regret bound is stronger. Have you thought about proving high-probability regret bounds?
-	Probably because I didn’t check the proof in appendix carefully, I personally don’t find the proof sketch in Section 5 to be very helpful. It is still hard for me to digest the high-level idea of the proof after reading it. Maybe the authors can think about reorganizing this part or cutting some part of this section (for example some detail about how to bound $F_t^\kappa$ term) and leave the space for other discussions.
-	Regarding the related literature, [1] and [2] are also quite related recent TS type algorithms. And currently there might be some technical issues in Xiong et al. [2021] as claimed by the authors.
-	It would be better to discuss the tightness of the bound when specialized to the linear MDP setting. I think there are a few results about the lower bound in linear MDP, e.g. [3].

[1] Agrawal, Priyank, Jinglin Chen, and Nan Jiang. "Improved Worst-Case Regret Bounds for Randomized Least-Squares Value Iteration." Proceedings of the AAAI Conference on Artificial Intelligence. Vol. 35. No. 8. 2021.
[2] Jafarnia-Jahromi, Mehdi, et al. "Online Learning for Stochastic Shortest Path Model via Posterior Sampling." arXiv preprint arXiv:2106.05335 (2021).
[3] Zhou, Huozhi, et al. "Nonstationary reinforcement learning with linear function approximation." arXiv preprint arXiv:2010.04244 (2020).

**Time Spent Reviewing:**

7

---

> ### Author Response · Authors · 2021-08-10
> **Reply to Reviewer f4DA**
>
> Thank you for your careful assessment and detailed feedback. We will include your comments in our next revision and respond to the main questions below:
>
> > I notice that the optimistic prior is not symmetric for all level h since it only includes $\exp⁡(\lambda f^1(x^1))$. Can you comment on why this special definition is helpful for the analysis?
>
> That is a great question. In essence, an optimistic prior at the initial observations is sufficient and, through the likelihood, propagates to later observations as well. We considered adding such a prior on observations seen at all time steps so far, but realized that this does not have the desired effect in certain cases. This happens when the function class negatively couples observations visited so far with observations that have not been visited yet, where such a prior may in fact promote pessimism.
>
> > Theorem 1 itself is hard to parse, since we generally don’t know how large the κ terms and the decompiling coefficient terms are. I think the magnitude of $\kappa$ is only investigated for the linear MDP case. Is it possible to obtain similar bounds of $\kappa$ for other settings?
>
> We will add further examples to the paper to help interpreting the general statement. For example, when the prior is uniform, then $\kappa$ in our result takes the role of the function class complexity ($\ln |F|$ or VC-dim) in OFU-based algorithms.
>
> > One limitation of the algorithm is the computationally inefficiency. Besides, I think the algorithm requires the knowledge of $\kappa$ and decompiling coefficient terms. Is this true? If so, is it possible to extend to the unknown case?
>
> Note that only the optimal choice for parameter $\lambda$ depends on $\kappa$, otherwise, our algorithm is independent of $\kappa$. For this reason, we state Theorem 1 for general $\lambda$ as well and our results suggest that the performance of our algorithm degrades gracefully with deviations of $\lambda$ from its optimal choice. Such a mild dependence on hyperparameters is common in existing TS and OFU algorithms and one can extend our algorithm using general model selection approaches (see e.g. [4] and references therein) to the case where $\kappa$ is unknown.
>
> > In Corollary 2, it seems that the priors are assumed to chosen uniformly so that a bound on $\kappa$ can be achieved. Is there any difficulty to extend it to more general priors?
>
> We assumed a uniform prior for convenience to achieve simple form for our regret bound but extensions to other, more general priors are possible.
>
> > The regret bounds are the expected regret bounds. It is generally believed that the high-probability regret bound is stronger. Have you thought about proving high-probability regret bounds?
>
> We chose to prove regret bounds in expectation since our analysis technique lends itself to those but we believe that high-probability bounds could be obtained as well with slightly more involved arguments.
>
> > [...] Maybe the authors can think about reorganizing this part [Section 5] or cutting some part of this section (for example some detail about how to bound Ftκ term) and leave the space for other discussions.
>
> Thank you for this suggestion. We will streamline the proof sketch accordingly.
>
> > Regarding the related literature, [1] and [2] are also quite related recent TS type algorithms. And currently there might be some technical issues in Xiong et al. [2021] as claimed by the authors.
>
> Thank you for pointing out those works. We will add a citation for [1] and [2] to the paper (and for Xiong et al as well if their technical issues will be resolved).
>
> [4] Cutkosky, Ashok, et al. "Dynamic balancing for model selection in bandits and rl." ICML 2021.

---

### Official Review · Reviewer_ESrm · 2021-07-17

**Rating:** 5
**Confidence:** 4

**Summary:**

The work proposes a new Thompson-sampling inspired exploration algorithm for exploration with general function approximation.

**Main Review:**

The paper is written clearly and nicely organized.

The TS-inspired algorithm examined in this work can function on most of the recently proposed settings, i.e., Bellman completeness, Bellman Eluder dimension etc.. A new complexity measure (Online decoupling coefficient) is introduced and connects nicely with other definitions.

A Thompson sampling algorithm, when designed correctly, has 3 major benefits beyond ofter superior empirical performance:

- it is computationally efficient, as it requires sampling instead of solving a potentially non convex optimization problem
- it avoids the construction of confidence intervals, which can be an issue with complex approximators
- a similar algorithmic structure generalizes across different settings

The current proposal here is not computationally efficient. In my opinion, this is a major dealbreaker.

To my understanding, all optimization based works (e.g., Zanette et al ‘20, Jin et al. '21 etc) that are computationally intractable admit an easy randomized version which is also computationally intractable: simply sample from the feasible set until a near-optimal solution is found. One can check, that a good enough solution is normally found after exponentially (in d,H for the simple linear case) many samples.

In this light, I'm not sure how this work differs from those that are intractable and do not explicitly introduce randomization; randomization here seems to be an ineffective way to solve a complex problem and does not bring advantages compared to non-randomized methods. I think the work looses the spirit of TS algorithms and does not make significant algorithmic or analytical contribution into the correct design and use of randomization to tackle complex reinforcement learning problems.


**Time Spent Reviewing:**

1

---

> ### Author Response · Authors · 2021-08-10
> **Reply to Reviewer ESrm**
>
> We thank the reviewer for their assessment and address the concerns about computational efficiency below.
>
> We believe that TS methods have potential computational advantages in practice and that’s why they are an important class of algorithms in practice. While there are no formal proofs of their computational efficiency, there are theoretical results suggesting that under suitable conditions, sampling can be computationally more efficient than optimization for nonconvex problems [1]. We will also give some additional justifications on the potential computational benefits below. Due to their practical values, we believe a good theoretical understanding of TS algorithms is a positive and meaningful contribution to the RL literature.
>
> When discussing computational efficiency of algorithms that can handle the rich observation setting, as ours does, it may be worth shedding light on what we can expect. The question of designing an algorithm that is provably statistically and computationally efficient in this setting is still an open and actively researched problem. A statistically efficient method that can be implemented exactly with polynomially many elementary operations would be great but may be too much to ask. Note that, for example, the computational goal for optimization-based algorithms is more modest: existing works aim to make them implementable with standard regression and classification oracles that may not be computationally efficient in general but are known to achieve good results empirically (e.g. cost-sensitive classification, see e.g. discussion in [2]). Whether algorithms exist that achieve even this more modest goal without making further assumptions (as in [3]) is still an open problem.
>
> We therefore would argue that designing a provably statistically efficient algorithm that relies on a different set of computational tools may still be a useful contribution regarding the computational question in the rich observation setting. In fact, our algorithm can leverage the large arsenal of sampling procedures (MCMC inference, Langevin dynamics etc.) developed in the Bayesian community, that have been shown to perform well empirically.
> A simple randomization of optimization-based algorithms by sampling from the feasible set is likely not going to be as computationally efficient as posterior sampling in practice since it does not incorporate the structure of the potentially very helpful prior and likelihood. If one does come up with an approach that does include them in the randomization, then the algorithm is arguably closer to TS than OFU and is better analyzed very differently from OFU, as we do here.
>
> That said, we would like to emphasize that the focus of this work was to study *statistical* and not computational efficiency. As the reviewer points out, there are several benefits of a TS algorithm, including: (1) avoid constructing explicit confidence intervals and (2) algorithm structure that works well in many settings. One could add that (3) TS algorithms can naturally incorporate prior information, which is particularly useful in multi-task settings. We would like to highlight that our algorithm does have these benefits and our work shows that there is no statistical gap between TS and OFU algorithms by proving the first matching regret bounds for TS in this general setting. We believe that these are significant contributions in their own right, independent of the computational question.
>
> * [1] Ma et al. "Sampling can be faster than optimization" Proceedings of the National Academy of Sciences 116 (42), 20881-20885
> * [2] Dann et al. "On oracle-efficient PAC RL with rich observations." NeuIPS 2018.
> * [3] Misra, Dipendra, et al. "Kinematic state abstraction and provably efficient rich-observation reinforcement learning." ICML 2020.

---

### Official Review · Reviewer_6qxa · 2021-07-18

**Rating:** 7
**Confidence:** 3

**Summary:**

The authors propose a posterior-sampling algorithm for RL on a family of MDPs and value-function classes. Although an efficient sampling procedure is not given, the authors derive regret bounds for this family of MDPs that match (or improve) the existing bounds in expectation.

**Limitations And Societal Impact:**

Yes

**Main Review:**

Overall, the presentation is clear and readable. Although the contribution is mainly theoretical and an efficient implementation of the algorithm is missing, I believe the work is significant enough for publication in NeurIPS.

I did not check the proofs but the results and the arguments seem believable. In comparison with existing results (e.g. UCB-LSVI), it should probably be noted that the regret bound for posterior-sampling presented here is in expectation whereas the bounds in many existing works are with high-probability.

Some minor glitches can be found here and there. I urge the authors to double-check everything. For example, in Line-69 P^h is defined for h=1...H whereas the definition of Q (Line-78) uses P^{h+1} so apparently P is needed for h=2....(H+1). This is distracting for the astute reader.

**Time Spent Reviewing:**

2

---

> ### Author Response · Authors · 2021-08-10
> **Reply to Reviewer 6qxa**
>
> We thank the reviewer for their assessment and feedback. We will make sure to carefully go through the paper and correct the minor glitches. Regarding the expectation vs high-probability bounds, we will add a note to the paper as well. We chose to prove regret bounds in expectation since our analysis technique lends itself to those but we believe that high-probability bounds can be obtained as well with slightly more involved arguments.

---

### Official Review · Reviewer_JTUW · 2021-07-21

**Rating:** 6
**Confidence:** 4

**Summary:**

This work introduces a conditional posterior sampling algorithm that is a model-free posterior sampling reinforcement learning algorithm and can be applied to general Markov decision processes and value function classes. Authors provide theoretical guarantees for the proposed algorithm.  Actually, it has been proved that the worst-case regret of the proposed algorithm is near-optimal and that improves the best known regret bounds for posterior sampling approaches (OPT-RLSVI, UCB-LSVI). The key component of the proposed algorithm is the definition of the likelihood over the collected samples (observations - state, action, reward) given a value function. To conclude, it is a theoretical work that shows that there is no statistical efficiency gap between OFU and posterior sampling algorithms.

**Limitations And Societal Impact:**

I think that the limitations of this work have been adequately addressed by the authors

**Main Review:**

**Strengths**
- This work introduces a conditional posterior sampling algorithm that is a model-free posterior sampling reinforcement learning algorithm that can be applied to general Markov decision processes.
- Theoretical guarantees for the proposed algorithm are provided. The worst-case regret of the proposed algorithm is near-optimal
- The worst-case regret of the algorithm improves the best known regret bounds for posterior sampling approaches, such as OPT-RLSVI, UCB-LSVI. Apart from that, it has been shown that a posterior sampling method can achieve the same guarantees with the OFU based algorithms.

**Weaknesses**

- The main limitation of this work is the lack of any empirical analysis. It would be of high importance for the reader to be able to check that the theoretical findings are aligned with those of the experiments (even on toy environments).
- Another open question is if the proposed algorithm is actually applicable in practice.  As mentioned by the authors, the sampling of the posterior is not computationally tractable in its current form (Eq. 3).

To conclude, despite the theoretical guarantees of the proposed conditional posterior sampling algorithm, authors should make clear if this algorithm could be also applied in practice.

**Minor comments/typos**
- l. 150 pairs. -> pair,
- l. 160: in comparison the the

**Time Spent Reviewing:**

7

---

> ### Author Response · Authors · 2021-08-10
> **Reply to Reviewer JTUW**
>
> Thank you for the comments. The primary focus of our work is the design of a model-free posterior sampling algorithm and its theoretical analysis. We agree that studying the performance of our algorithm empirically is an exciting direction and we plan to carry out a careful experimental comparison in future work. We believe that our theoretical contributions, including the proof that TS is as statistically efficient as optimism-based algorithms, are significant on their own and would find it very difficult to present them and the results of a careful empirical evaluation in sufficient detail in the same paper.
>
> Although we leave an empirical study to future work, we strongly believe that our algorithm can be implemented practically using the many techniques developed in the Bayesian community to sample approximately from the posterior (e.g. stochastic gradient Langevin dynamics).
>
> Thank you for pointing out the typos. We will make sure to fix them.

---

> > ### Comment · Reviewer_JTUW · 2021-09-12
> > **Acknowledge of reading authors' response**
> >
> > I would like to thank the authors for their response.
> >
> > Having read the other reviewers and the authors' rebuttal, I have decided to keep my initial rating (borderline). Some issues have also raised about the contribution of this work by the third reviewer.

---

### Decision · Program_Chairs · 2021-09-27

**Decision:**

Accept (Poster)

**Comment:**

This is a well-written and interesting paper.  However, as multiple reviewers pointed out, its not clear whether the approach considered is computationally viable.  As one of the reviewers suggests, this could be a dealbreaker.  At the same time, I do appreciate that the work in sorting out sample complexity for value function sampling under these idealized conditions.  After thinking about this some, I am inclined to recommend accepted.